JCB Journal of Cell Biology

# Tejas functions as a core component in nuage assembly and precursor processing in *Drosophila* piRNA biogenesis

Yuxuan Lin[1], Ritsuko Suyama[1], Shinichi Kawaguchi[1], Taichiro Iki[1], and Toshie Kai[1]

**PIWI-interacting RNAs (piRNAs), which protect genome from the attack by transposons, are produced and amplified in membraneless granules called nuage. In *Drosophila*, PIWI family proteins, Tudor-domain-containing (Tdrd) proteins, and RNA helicases are assembled and form nuage to ensure piRNA production. However, the molecular functions of the Tdrd protein Tejas (Tej) in piRNA biogenesis remain unknown. Here, we conduct a detailed analysis of the subcellular localization of fluorescently tagged nuage proteins and behavior of piRNA precursors. Our results demonstrate that Tej functions as a core component that recruits Vasa (Vas) and Spindle-E (Spn-E) into nuage granules through distinct motifs, thereby assembling nuage and engaging precursors for further processing. Our study also reveals that the low-complexity region of Tej regulates the mobility of Vas. Based on these results, we propose that Tej plays a pivotal role in piRNA precursor processing by assembling Vas and Spn-E into nuage and modulating the mobility of nuage components.**

## Introduction

Transposons (transposable elements, TEs) are mobile genetic elements that exist in the genomes of all eukaryotic organisms and they occupy a substantial portion of genomes (Huang et al., 2012; Samarasinghe et al., 2017). They directly impair genomes by causing double-strand breaks, promoting ectopic recombination, and abolishing gene expression (Hedges and Belancio, 2011; Hedges and Deininger, 2007). PIWI-interacting RNAs (piRNAs), a class of 23–29-nt gonad-specific small RNAs, protect genome integrity by mitigating any catastrophes in germline cells that will be transmitted to the next generations (Brennecke et al., 2007; Cenik and Zamore, 2011; Czech and Hannon, 2011). piRNAs are quite conserved and widely found among animals, and the model animal system, *Drosophila*, has been used to investigate and dissect the molecular mechanisms of piRNAs (Aravin et al., 2006; Girard et al., 2006; Grivna et al., 2006; Hirano et al., 2014; Lim and Kai, 2015).

*Drosophila* piRNAs are processed from long piRNA precursor transcripts derived from genomic loci called piRNA clusters, where inactive or fragmented transposons are deposited (Brennecke et al., 2007, 2008; Khurana et al., 2011; Malone and Hannon, 2009; Ozata et al., 2019; Rozhkov et al., 2013). Discrete piRNA clusters are active in gonads, where they produce dual-strand piRNA precursors in germline cells or unistrand piRNA precursors in somatic gonadal cells (Gleason et al., 2018; Ozata et al., 2019). In germline cells, nascent piRNA precursors are

transported to a unique, germline-specific membraneless structure called nuage in the perinuclear region via the Nxf3–Nxt1 pathway (ElMaghraby et al., 2019; Kneuss et al., 2019; Mendel and Pillai, 2019). Nuage consists of precursors and transposon RNAs being processed, two PIWI family proteins—Aub and Ago3—and other relevant components, DEAD-box RNA helicase Vasa (Vas), DEAH box helicase RNA helicase Spindle-E (Spn-E), and a group of Tudor domain-containing proteins (Tdrds), Krimper (Krimp), Tejas (Tej), Tudor, Tapas (Tap), Qin/Kumo, and Vreteno (Anand and Kai, 2012; Brennecke et al., 2007; Gillespie and Berg, 1995; Golumbeski et al., 1991; Gunawardane et al., 2007; Liang et al., 1994; Lim and Kai, 2007; Patil et al., 2014; Patil and Kai, 2010; Zamparini et al., 2011). After loading long piRNA precursors and transposon RNAs onto Aub and Ago3, they are cleaved and sliced into mature piRNAs, leading to the formation of antisense and sense piRNAs with a 10-nt complementarity (Aravin et al., 2007; Grimson et al., 2008; Houwing et al., 2007; Lim et al., 2014). These processed piRNAs are further amplified in nuage in a feed-forward amplification cycle called the ping-pong cycle. However, the molecular mechanisms of nuage assembly are still unclear.

Although Tdrds are multifunctional, their overall activities are not fully understood. They interact with symmetrically demethylated arginine (sDMA), which is usually present at the N-terminus of PIWI family proteins (Kirino et al., 2009, 2010;

[1]Graduate School of Frontier Biosciences, Osaka University, Osaka, Japan.

Correspondence to Toshie Kai: kai.toshie.fbs@osaka-u.ac.jp.



Li et al., 2009; Selenko et al., 2001), through the Tudor domain, thereby promoting aggregate formation in mammalian cells (Courchaine et al., 2021). This behavior implies the importance of molecular associations of Tdrds for nuage formation. Membraneless organelles composed of RNA and proteins are responsible for diverse RNA processing, including P-body and Yb body in *Drosophila*, which modulate the molecular organization in a process called phase separation (Hirakata et al., 2019; Kistler et al., 2018; Sankaranarayanan et al., 2021). Two Tdrds localized in *Drosophila* nuage—Tej and Tap—contain an extended Tudor domain (eTudor) and an additional Lotus domain that is conserved from bacteria to eukaryotes (Kubíková et al., 2021). The Lotus domain was previously reported to interact with Vas, which is required for the piRNA pathway (Jeske et al., 2015, 2017).

Of these two proteins, Tej/Tdrd5 is one of the key factors in the piRNA pathway in both *Drosophila* and mice (Patil et al., 2014; Patil and Kai, 2010; Yabuta et al., 2011). piRNAs are massively reduced with the displacement of other components from nuage in the absence of Tej/Tdrd5; however, the molecular functions of Tej remain elusive. Here, we identified the domains of Tej that interact with Vas and Spn-E, which are required for proper nuage formation and piRNA precursor processing, in addition to the contribution of the intrinsically disordered region (IDR) to the dynamics of other nuage components. We propose that Tej plays a pivotal role in piRNA precursor processing by recruiting Vas and Spn-E for nuage and modulating their dynamics for nuage assembly.

## Results

### Tej associates with Vas and Spn-E to form perinuclear nuage granules

To demonstrate the general assembly of nuage, we revisited the subcellular localization of the nuage components and dissected the detailed molecular mechanisms involving Tej (Gratz et al., 2013) using fluorescent-tagged Tej, Spn-E, and Ago3, as well as Vas and Aub. In particular, the expression and subcellular localization of these molecules in the ovaries were examined (Fig. 1 A; and Fig. S1, A and B; Kina et al., 2019). We detected a robust and tight colocalization of Tej-GFP with Vas-mCherry and Spn-E-mKate2 (mK2) in perinuclear nuage granules by super-resolution confocal microscopy (Fig. 1, A and B; Patil and Kai, 2010). The localization of GFP-Aub, mk2-Ago3, Vas-GFP, and Spn-E-mK2 was affected in *tej* mutant germline cells (Fig. 1, B and C; and Fig. S1 B; Patil and Kai, 2010). Vas-GFP was found as a smooth layer in the perinuclear region while Spn-E-mK2 was predominantly found in the nucleus of *tej* mutant germline cells, indicating that the colocalization of Vas and Spn-E was affected by the absence of *tej* (Fig. 1 C and Fig. S1 C).

The different subcellular localizations of Vas and Spn-E in the absence of *tej* prompted us to examine the physical interactions between Tej and Spn-E or Vas in vivo by crosslinking immunoprecipitation (CL-IP; Fig. 1 D). Using the ovarian lysate expressing GFP-Tej, both Spn-E and Vas and other major nuage components, such as Ago3, Aub, and Piwi, but not Ago2, were immunoprecipitated with Tej (Fig. 1 D; Saito et al., 2006; Wei

et al., 2012; Iki et al., 2020). The opposite direction of CL-IP for Vas-GFP or Spn-E-mK2 demonstrated the association of these components with Tej and the abovementioned nuage components but not with the reciprocal RNA helicases. Moreover, Vas and Spn-E were hardly immunoprecipitated by each other (Fig. 1 D). Moreover, Tej and Spn-E remained in nuage granules in the absence of Vas in the early-stage egg chambers where nuage was stably formed, whereas Tej and Vas were displaced from the perinuclear nuage without Spn-E (Fig. S1 D). These results suggest that Tej interacts with Vas and Spn-E in a different subcompartment.

### Tej interacts with Vas and Spn-E through distinct domains

We further dissected the interactions of Tej with Vas and Spn-E in S2 cells (Fig. 2, A and B). Following single transfection, GFP-Vas formed heterogeneous aggregates and mK2-Tej formed granules in the cytoplasm (Fig. 2 B), while GFP-Spn-E was mostly dispersed in the nucleus. Interestingly, co-expression with full-length Tej (Tej-FL) changed the localization of Vas or Spn-E; the cytoplasmic Vas or nuclear Spn-E was recruited into large cytoplasmic granules with Tej (Fig. 2 B). These results indicated that Tej can aggregate with cytoplasmic Vas and recruit nuclear Spn-E into cytoplasmic granules.

To identify the domains responsible for interacting with Vas and Spn-E, we generated truncated variants of Tej and examined their individual capability to recruit Vas and Spn-E (Fig. 2, A and B). Tej lacking the Lotus domain (Tej-ΔLotus) formed cytoplasmic aggregates and exhibited co-localization with Spn-E (Fig. 2 B). In contrast, consistent with the previous finding (Jeske et al., 2017), Tej-ΔLotus did not colocalize with Vas, indicating that the interaction between Tej and Vas depends on the Lotus domain. However, the eTudor domain-deleted Tej (Tej-ΔeTudor) was dispersed in the cytoplasm and similarly distributed with both Vas and Spn-E, suggesting that Tej-ΔeTudor recruits Spn-E to the cytoplasm (Fig. 2 B). Stepwise deletion from the N-terminus of Tej revealed that GFP-Spn-E aggregated into the cytoplasm with Tej variants other than Tej-Δ1–362, whereas only GFP was found in both the nucleus and cytoplasm (Fig. S2 A). These results indicate that 295–362 aa of Tej are essential for the recruitment of Spn-E to the cytoplasm. Further examination of the region of Tej 101–362 showed that the deletion of 338–362 aa of Tej (Tej 101–337) remarkably impaired the recruitment of Spn-E; however, the deletion of 350–362 aa (Tej 101–349) and Tej 101–362 did not produce such impairment (Fig. S2 B). Additionally, the individual substitutions of amino acids in the Tej 338–349 aa region also showed a weaker accumulation of Spn-E in the nucleus (Fig. S2 B). Thus, the domain involving 338–349 aa of Tej is critical for recruiting Spn-E and thus is referred to as "Spn-E Recruit Site" (SRS), which is highly conserved in *Drosophila* and vertebrates (Fig. S2 C).

The predicted structure of Spn-E and Tej, by AlphaFold v2.2, revealed that the C-terminus of the conserved helicase domain of Spn-E (334–393 aa) would be an interacting region for Tej (Fig. S2, D and E; and Fig. S3, A and B; Jumper et al., 2021). Spn-E$^{mut\ IN}$ has mutations in this region and was predominantly localized in the nucleus upon co-expression of Tej-FL or Tej-ΔSRS in S2 cells (Fig. 2 B and Fig. S2 F), suggesting that the

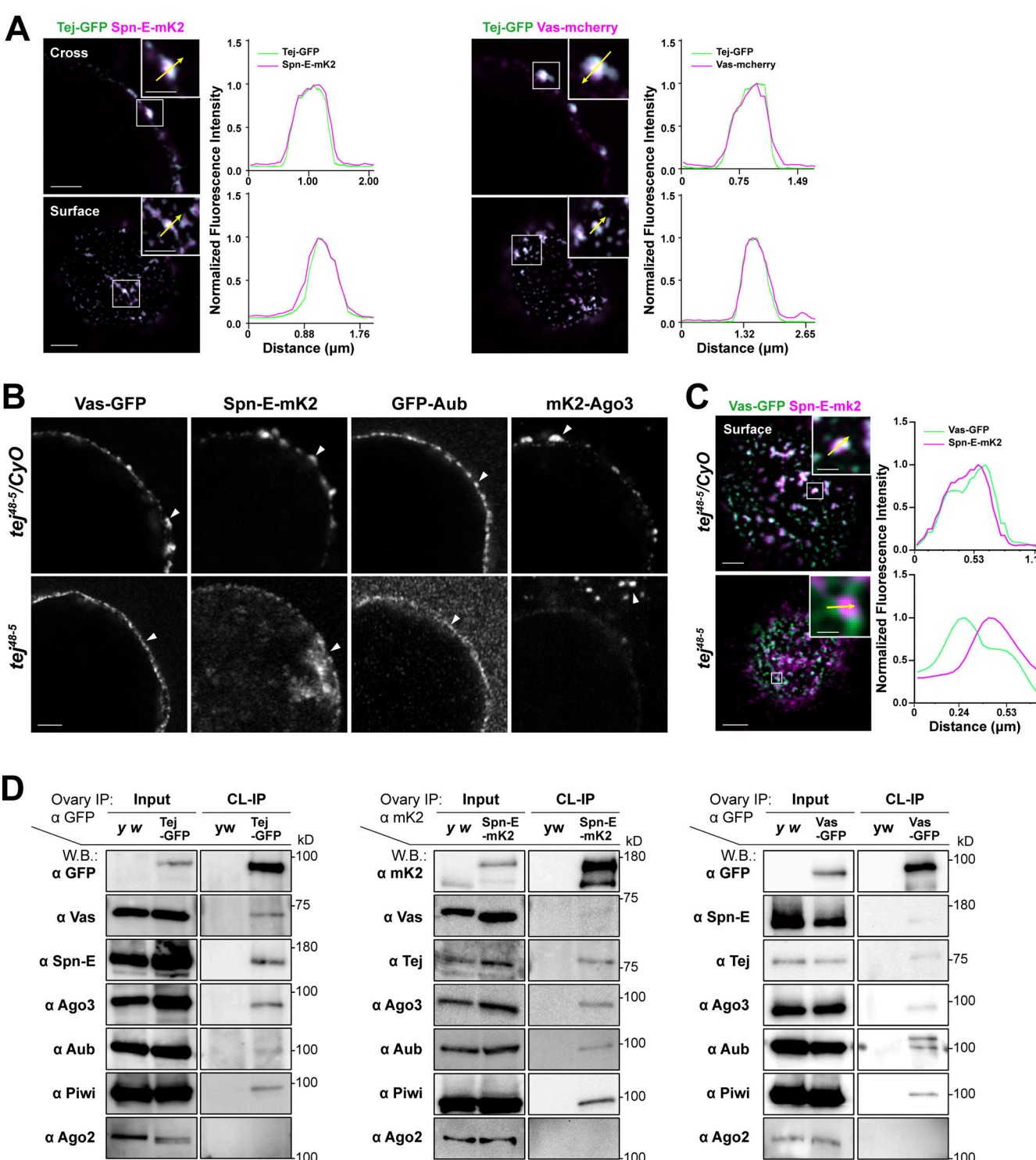

Figure 1. **Tej associates with nuage components and is required for proper nuage assembly. (A)** Colocalization of fluorescent-tagged endogenous Tej-GFP (green) and Spn-E-mK2 or Vas-mCherry (magenta) are shown in the cross-section of the nuclei (top panels) and the surface (bottom panels). The fluorescence intensity along the designated lines (yellow arrow in inset) is normalized to the highest value and plotted (right panels; $n \geq 3$, number of analyzed nuclei). **(B and C)** Tej is required for proper nuage formation. **(B)** The localization of Vas, Spn-E, Aub, and Ago3 is observed in the control ovaries ($tej^{48-5}/CyO$; top panels, white arrowheads) and $tej$ mutant ovaries ($tej^{48-5}$; bottom panels, arrowheads). **(C)** The localization of Vas-GFP (green) and Spn-E-mK2 (magenta) is observed in the control ovary ($tej^{48-5}/CyO$, top panel) and $tej$ mutant ovaries ($tej^{48-5}$; bottom panel). The fluorescence intensity along the designated lines (yellow arrow in inset) is normalized to the highest value and plotted (right panels; $n \geq 4$, number of analyzed nuclei). **(D)** Vas and Spn-E associate with Tej. Immunoprecipitants from the ovaries expressing Tej-GFP, Vas-GFP, or Spn-E-mK2 were analyzed using Western blotting. For the major piRNA biogenesis factors, namely Tej, Vas, Spn-E, Ago3, Aub, and PIWI, and Ago2 (an irrelevant siRNA component) were examined. Scale bars, 0.5 µm (inset of A upper panels), 1 µm (A upper panels, B, inset of A lower panels), 2 µm (A lower panels, C), 0.8 µm (inset of C upper panel), and 0.4 µm (inset of C lower panel). Source data are available for this figure: SourceData F1.

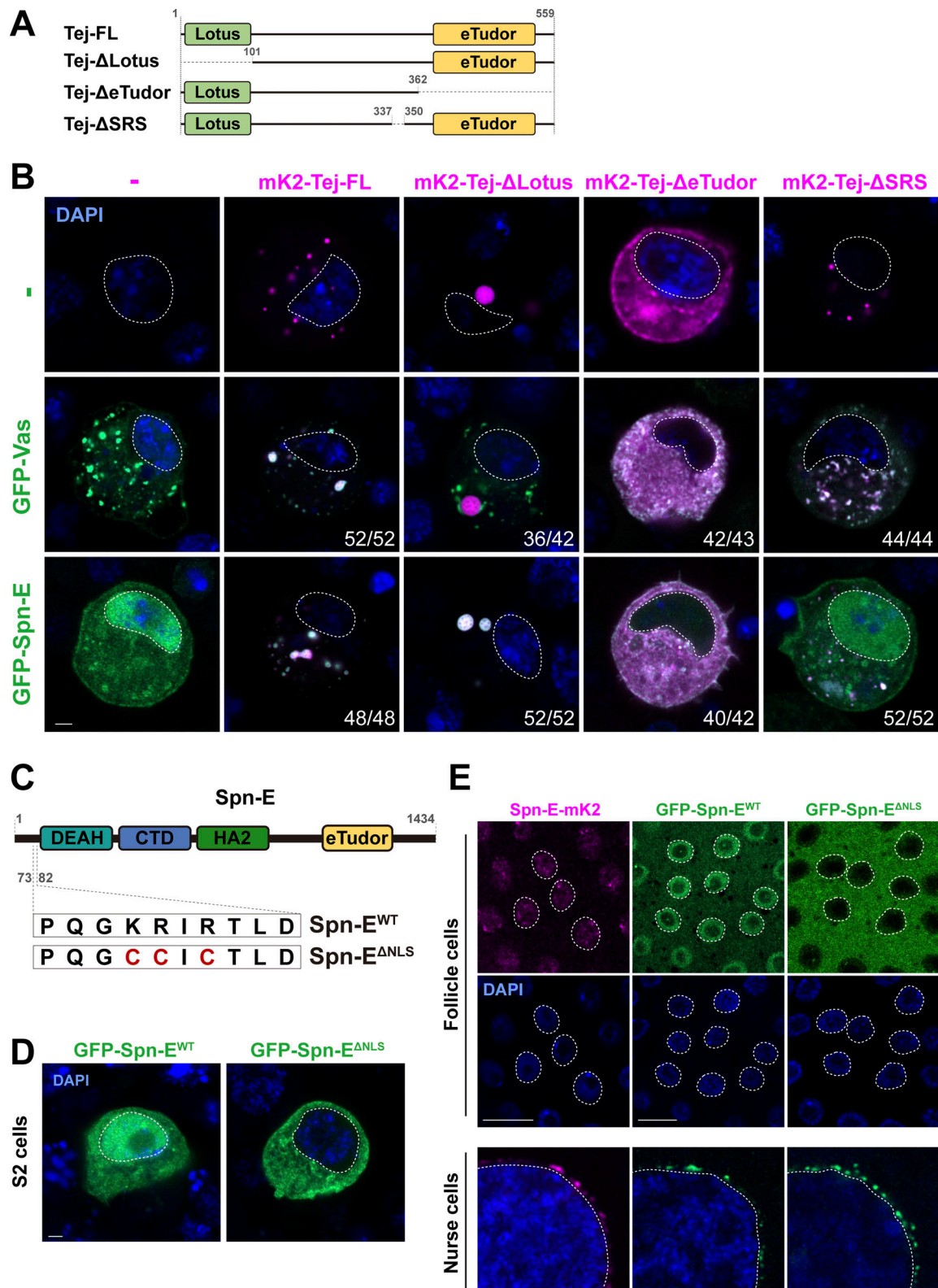

Figure 2. **Tejas recruits Vas and Spn-E via individual unique motifs. (A)** Schematic representation of the Tej full-length protein and the truncated variants expressed in the S2 cells: ΔLotus (deletion 1–100 aa), ΔeTudor (deletion 363–559 aa), ΔSRS (deletion 338–349 aa). **(B)** Tej recruits Vas and Spn-E via distinct domains. GFP-Vas or GFP-Spn-E (green, middle, and bottom panel, respectively) is coexpressed in S2 cells with mK2-Tej-FL or its truncated variants (magenta), Tej-ΔLotus, Tej-ΔeTudor, or Tej-ΔSRS. The predominant localization of these proteins and the number of cells displaying such patterns are shown. **(C)** Schematic representation of amino acid substitutions in Spn-E NLS (ΔNLS). The lysine at position 76 and arginines at positions 77 and 79 are substituted for cysteines. **(D)** GFP-tagged wild-type Spn-E (Spn-E^WT, green, left) and NLS-deleted Spn-E (Spn-E^ΔNLS, green, right) are expressed in S2 cells. **(E)** Spn-E-mK2

shows distinct localizations in ovarian somatic cells and nurse cells (magenta, left panels). GFP-tagged Spn-E$^{WT}$ or Spn-E$^{\Delta NLS}$ is expressed either by a somatic driver, *tj-Gal4*, or a germline driver, *Nos-Gal4*, in *spn-E* mutant ovaries (*spn-E$^{616}$/Df*). The nuclei are stained with DAPI (blue) and denoted with dotted circles in B, D, and E. Scale bars, 2 µm (B and D), 10 µm (E upper panels), 5 µm (E lower panels).

loss of their interaction perturbed the recruitment of SpnE into the cytoplasm in S2 cells. Moreover, mutations of a newly identified potential class II monopartite nuclear localization signal (NLS) in the N-terminal Spn-E (Fig. 2 C; Kosugi et al., 2009) excluded the nuclear localization of S2 cells (Fig. 2 D). Similarly, both endogenous Spn-E-mK2 and transgenic GFP-Spn-E were found in the nucleus of wild-type follicle cells, whereas they were found in nuage in germline cells (Fig. 2 E). Furthermore, Spn-E DNLS did not change nuage localization in germline cells, whereas it localized to the cytoplasm of the follicle cells (Fig. 2 E). Collectively, these results suggest that the Tej-Spn-E interaction requires both SRS of Tej and the predicted interface on Spn-E, thus leading to the localization of Spn-E in nuage.

**Tej functions in the proper processing of piRNA precursors**
To explore the role of Tej in piRNA biogenesis, we analyzed 23–29-nt-small RNAs bound to Aub or mK2-Ago3. The amount of total piRNAs bound to Aub or mK2-Ago3 was markedly reduced in *tej* mutant ovaries compared with that in the heterozygous control (Fig. S4 A and B). Consistently, Aub- or Ago3-bound piRNAs mapped to the genomic regions *38C* or *42AB* were remarkably reduced in *tej* mutant ovaries (Fig. 3 A). Moreover, the 1U and 10A preferences of Aub-bound antisense or Ago3-bound sense piRNAs were notably abolished (Fig. 3 B). Concomitantly, cluster transcripts derived from *38C* or *42AB* were upregulated in the *tej* mutant, whereas *flamenco*, an ovarian somatic piRNA precursor, was not affected (Fig. 3 C). Other nuage component mutants, such as *vas*, *spn-E*, and *krimp*, also exhibited similar defects, whereas *nxf3* did not (Fig. 3 C).

We then employed high-resolution hairpin chain reaction in situ hybridization (HCR-FISH; Choi et al., 2018) to examine the subcellular localization of piRNA precursors and found that the foci of the cluster transcripts of *38C* or *42AB* were barely detectable in the control (Fig. 3 D). By contrast, the foci accumulated more in the vicinity of the perinuclear region in the *tej*, *spn-E*, and *vas* mutant germlines (Fig. 3, D and E, 40–60% accumulation in the mutants and 18–26% in the control); however, no accumulation was discernible in the *nxf3* mutant as previously reported (ElMaghraby et al., 2019; Fig. 3 E, ~10%) or the *krimp* mutant (Fig. 3, D and E, ~28%; Kneuss et al., 2019). In conclusion, these results suggest that precursor processing into mature piRNAs was affected by the loss of Tej or interacting components Vas and Spn-E, thus leading to their accumulation around the perinuclear region.

**Tej functions in the processing of piRNA precursors via Vas and Spn-E recruitment to perinuclear nuage granules in vivo**
To address how the Lotus domain and SRS of Tej coordinate the recruitment of Vas and Spn-E, respectively, we expressed GFP-Tej variants in *tej* mutant germline cells with Vas-mCherry or Spn-E-mK2 (Fig. 4, A and B; and Fig. S5 A). Tej-FL and Tej-ΔLotus were observed as granules in the perinuclear region of

the germline cells (Fig. 4 B; Patil and Kai, 2010). GFP-Tej-FL recruited both Vas and Spn-E to perinuclear nuage granules; Tej-ΔLotus significantly segregated Vas foci from the nuage granules, with a thin layer spread along the perinuclear region; however, the Spn-E localization did not change (Fig. 4 B). Surprisingly, Tej-ΔSRS did not affect the localization of Spn-E or Vas in nuage granules in vivo, which is inconsistent with the observation in S2 cells (Fig. 2 B and Fig. S5 A). Thus, we further deleted the regions containing highly conserved residues among other species in addition to SRS (Tej-ΔeSRS, devoid of 329–349 aa; Fig. S2 C; and Fig. 4, A and B) and found that Tej-ΔeSRS remained in granularized nuage in the perinuclear region, which is similar to the case of Tej-FL. Interestingly, the majority of Spn-E was localized to the nucleus, while Vas predominantly remained in nuage although it was partially displaced (Fig. 4 B). In contrast, Tej-ΔeTudor lost its granular formation and was distributed in both perinuclear and cytoplasmic regions and colocalized partly with Vas and only a small amount of Spn-E localized around the perinuclear region (Fig. 4 B). Since Vas and Spn-E exhibited comparable expression in all ovaries expressing Tej variants (Fig. S5 B), ectopic localization of Vas and Spn-E was attributed to the molecular features of Tej variants. These microscopic observations were further supported by a CL-IP assay of ovarian lysates of each Tej variant. Unlike Tej-FL, Tej-ΔLotus lost its interaction with Vas but maintained it with Spn-E, while Tej-ΔeSRS interacted with Vas but not with Spn-E (Fig. S5 C). Moreover, Tej-ΔeTudor interacted with Vas and marginally with Spn-E. In conclusion, our results suggest that specific domains of Tej contribute to the proper formation of nuage granules and engage Vas and Spn-E in piRNA biogenesis in vivo.

We further examined the localization of piRNA cluster transcripts upon the expression of each Tej variant in *tej* mutant germline cells and quantified the foci of the cluster transcripts of *38C* or *42AB* in the perinuclear region (Fig. 4 C). Tej-FL rescued the accumulation of the precursor transcripts from *38C* and *42AB* around the perinuclear region of *tej* mutants, which was similar to that of the *y w* (Fig. 3 D and Fig. 4 C). Notably, Tej-ΔLotus still accumulated precursors that were concentrated around partially assembled nuage granules. Similarly, Tej-ΔeSRS showed a lower accumulation of foci around the malformed nuage, and more foci were scattered in the perinuclear region (Fig. 4 C). In contrast, Tej-ΔeTudor barely accumulated the precursor foci in the perinuclear region (Fig. 4 C). qRT-PCR showed that Tej-ΔLotus or Tej-ΔeSRS in *tej* mutants upregulated the cluster transcripts while Tej-ΔeTudor resulted in a milder upregulation (Fig. 4 E). Tej-FL almost completely rescued the upregulation of *HeT-A* and expression of other transposons, whereas Tej-ΔLotus and Tej-ΔeSRS could not rescue it (Fig. 4, D and F; Piñeyro et al., 2011). Tej-ΔeTudor did, however, rescue their derepression, albeit to a milder extent, than the other variants (Fig. 4, D and F). In conclusion, our results suggest that the function of Tej in the processing of piRNA precursors is to repress transposons through

Figure 3. **Perturbation of piRNA precursor processing collapses piRNA biogenesis in *tej* mutant ovaries. (A)** The Aub- (left panels) and Ago3-bound (right panels) small RNAs in the control and *tej* mutant ovaries are mapped to the major piRNA clusters, *38C* and *42AB*. Sense (blue) and antisense (red) piRNAs are indicated by means of upward and downward peaks, respectively. The gray bars indicate the independent biological replicates. **(B)** Nucleotide bias of the transposon-mappable Aub- and Ago3-bound piRNAs in the control and *tej* mutant ovaries. piRNA reads are plotted as sequence logos. **(C–E)** Cluster *38C*- and *42AB*-derived piRNA precursors are accumulated in the mutant ovaries of piRNA pathway components, *tej*, *spn-E*, *vas*, *krimp*, and *nxf3*. **(C)** Fold changes of the piRNA precursors, *cluster 38C*, *42AB*, and *flam*, in the mutant ovaries. Error bars indicate standard deviation (*n* = 3, number of analyzed independent experiments). **(D)** piRNA precursors are detected in the control (*y w*) and mutant ovaries of the indicated genotypes using HCR-FISH (green; Table S1). The nuclear envelope is stained by WGA (pseudo-white), and the nuclear DNA is stained with DAPI (blue). Scale bar, 1 μm. **(E)** The ratio of the fluorescence intensities of the piRNA precursors in the nuclear membrane vicinity of *tej*, *spn-E*, *vas*, *krimp*, and *nxf3* mutant germline cells. The signal intensity of the foci located inside and outside the nuclear membrane within a distance of 5% of the nucleus diameter is quantified (*n* = 10, number of analyzed nuclei), normalized with that in the nucleus, and plotted as percentiles relative to the total intensity. The numbers on the bars denote percentiles.

the proper recruitment of Vas and Spn-E to perinuclear nuage granules.

### Tej IDR enhances the mobility of Vas in nuage
Tej induced the formation of granular-like aggregates in S2 cells (Fig. 2 B, Fig. S2 A, and Fig. S6 A), implying a role of Tej in forming the membraneless organelle, nuage. Tej contains a predicated intrinsically disordered region (Dosztányi et al., 2005; Mészáros et al., 2018; Fig. 5 A), prompting us to examine its contribution to condensate formation. We investigated

the aggregate formation of Tej-Vas-Spn-E by cotransfection in S2 cells. They formed core-shell granules with Spn-E concentrated in the center, and Vas and Tej located toward the periphery (Fig. 5 B). We treated the cells with 1,6-hexanediol (1,6-HD) to disturb the weak hydrophobic interactions and found that Vas significantly relocalized from the periphery to the center of the granule together with Spn-E while Tej remained at the periphery (Fig. 5 B). This finding suggests that the higher mobility of Vas was intervened by the weak hydrophobic interactions among these proteins.

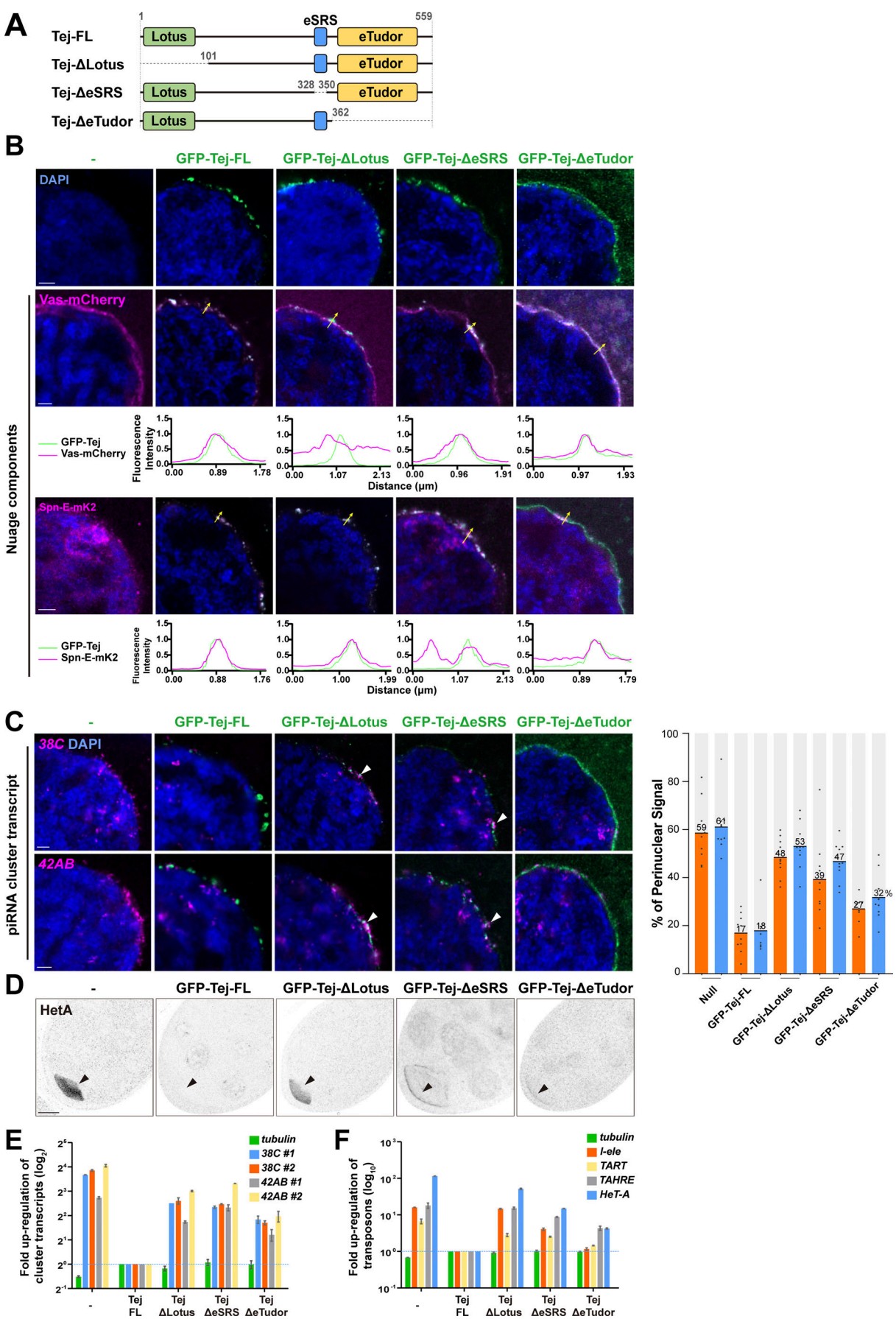

**Figure 4. Recruitment of Tej and Spn-E to the perinuclear nuage is required for the proper processing of piRNA cluster transcripts and TE repression. (A)** Schematic representation of GFP-Tej truncated variants expressed in germline cells. **(B)** Vas-mCherry (magenta, middle panels) or Spn-E-mK2 (magenta, bottom panels) are observed in *tej* mutant germline cells (*tej*[48−5]) expressing the GFP-tagged transgenes, full-length Tej (Tej-FL), Tej-ΔLotus, Tej-ΔeSRS or Tej-ΔeTudor (green, top panels). The yellow arrows indicate the perinuclear aggregations. The fluorescence intensity along the designated lines (yellow arrow) is normalized to the highest value and plotted (*n* ≥ 3, number of analyzed nuclei). **(C)** HCR in situ hybridization showing piRNA precursors derived from clusters *38C* and *42AB* (magenta) in *tej* mutant germline cells (*tej*[48−5]) and those expressing Tej variants. The white arrowheads indicate precursors accumulated in the proximity of the perinuclear nuage granules. DNA is stained with DAPI (blue). Scale bars, 1 μm (B and C; *n* ≥ 3, number of analyzed nuclei). The ratio of the fluorescence intensities of the piRNA precursors in the nuclear membrane vicinity of Tej variants rescued germline cells. The signal intensity of the foci located inside and outside the nuclear membrane within a distance of 5% of the nucleus diameter is quantified (*n* = 10, number of analyzed nuclei), normalized with that in the nucleus, and plotted as percentiles relative to the total intensity. The numbers on the bars denote percentiles. **(D)** Immunostaining of HeT-A Gap protein in *tej* mutant germline cells (*tej*[48−5]) and those expressing Tej variants. The black arrows indicate the accumulation of HeT-A in the oocytes. **(E and F)** The fold changes of transcripts derived from piRNA clusters *38C* and *42AB* (E) and those of *I-element, TART, TAHRE,* and *HeT-A* (Klenov et al., 2011; Savitsky et al., 2006; Chen et al., 2016) (F) with *tubulin* as a control. All values are normalized to *rp49* and shown as a comparison to the expression level in the presence of Tej-FL. Error bars indicate standard deviation (*n* = 3, number of analyzed independent experiments).

We confirmed that cytoplasmic condensates were formed irrespective of the fluorophore and their location (Fig. 2 B, Fig. S2 A, and Fig. S6 A) and examined the contribution of each domain of Tej toward the cytoplasmic aggregates. Tej-ΔLotus displayed spherical aggregates, whereas the eTudor domain (Tej-Δ1–362) was sufficient to form amorphous aggregates (Fig. 2 B and Fig. S6 A). In contrast, Tej-101–362 was broadly distributed in the cytoplasm (Fig. S6 A), indicating that the eTudor domain contributes to the condensate formation with the help of the middle part of Tej, while the Lotus domain does not. Notably, Tej-ΔIDR retained the ability to recruit both Vas and Spn-E (Fig. 5 C), suggesting that the IDR region of Tej appears to be dispensable for the recruitment of both.

To explore the contribution of the IDR region and its role in piRNA biogenesis via LLPS, we analyzed the dynamics of Tej, Spn-E, and Vas in S2 cells using fluorescence recovery after photobleaching (FRAP). Upon photobleaching, the fluorescence intensity of GFP-Tej-FL recovered rapidly (Fig. S6 B). In contrast, Tej-ΔIDR showed a significantly lower recovery rate, whereas Tej-ΔSRS remained relatively stable. Notably, Tej-ΔLotus recovered faster than Tej-FL, probably due to the loss of the Lotus domain allowing the exposure of the flexible IDR (Fig. S6 B). The high dynamics of Tej suggested that Tej may contribute to the dynamics of the other nuage components, possibly via the IDR. FRAP experiments showed that more than 80% of Vas was recovered within 90 s with Tej-FL ($t_{1/2}$: 23.16 ± 11.46 s); however <25% was slowly recovered with Tej-ΔIDR ($t_{1/2}$: 59.28 ± 40.73 s), indicating that Tej IDR facilitated the mobility of Vas (Fig. 5 D, upper panel). However, <40% of Spn-E was slowly recovered with Tej-FL or Tej-ΔIDR (Fig. 5 D, lower panel), indicating Spn-E formed rather static granules.

Finally, we investigated the potential function of Tej IDR in vivo. In *tej* mutant ovaries, Tej-ΔIDR formed condensed granules similar to the case of Tej-FL in the perinuclear region (Fig. 5 E and Fig. S6 C). However, we observed a slight upregulation of some transposons and HeT-A (Fig. 5, E and F) by qRT-PCR and immunostaining. Next, we examined Vas mobility in vivo because of the higher mobility compared with Spn-E in S2 cells (Fig. 5 D and Fig. S6 D). Notably, the mobility of Vas in nuage was more attenuated by Tej-ΔIDR than by Tej-FL (recovery rate: 82%, $t_{1/2}$: 10.79 s with Tej-FL vs. 62%, $t_{1/2}$: 77.35 s with Tej-ΔIDR; Fig. 5 G). In conclusion, these results suggest that

Tej IDR contributes to the mobility of nuage components, not only that of Tej itself but also that of Vas, while Spn-E remains relatively static.

## Discussion

The piRNAs in *Drosophila* germline cells are produced and amplified in the membraneless organelle, nuage, which is assembled by orderly recruitment of the corresponding components to ensure its proper function. Although its precise function has not been clarified, our findings demonstrate that Tej plays a crucial role in recruiting RNA helicases Vas and Spn-E to nuage through distinct domains, namely, Lotus and SRS. Our results provide new insights into the regulation of stepwise piRNA precursor processing by Tej, Spn-E, and Vas in the initial phase of piRNA biogenesis prior to the ping-pong amplification cycle. Tej recruits these helicases for the engagement of the precursors involved in further processing of nuage, thereby also controlling the dynamics of these nuage components (Fig. 6).

Our results confirmed that the Tej Lotus domain recruited Vas to nuage, which is consistent with the fact that it enables Vas to hydrolyze ATP for RNA release (Jeske et al., 2017). We newly identified that the SRS motif in Tej is responsible for Spn-E recruitment to nuage. Full deletion or single amino acid substitution of SRS significantly disrupted Spn-E recruitment to Tej granules in S2 cells, whereas further deletions of eight amino acids other than SRS, eSRS, were critical for recruiting Spn-E to nuage in the ovaries. This result raises a possibility that Tej, as well as other factors, may assist the recruitment of Spn-E to nuage in the ovaries. Another protein known as Tap, which is a fly counterpart of TDRD7 and harbors Lotus and eTudor domains, has previously been reported to participate in the piRNA pathway and interact with Vas (Jeske et al., 2017; Patil et al., 2014). However, since Tap lacks the SRS found in Tej, it is unlikely to be involved in the recruitment of Spn-E. The mouse homolog of Spn-E (TDRD9) is localized in both nuage and the nucleus in prespermatogonia (Shoji et al., 2009; Wenda et al., 2017), and might perform different functions that remain elusive. Our finding suggests a possibility that the intrinsically nuclear protein Spn-E was deliberately recruited to nuage via Tej to exert a unique function, such as piRNA precursor processing. In contrast, the eTudor domain mainly contributes to

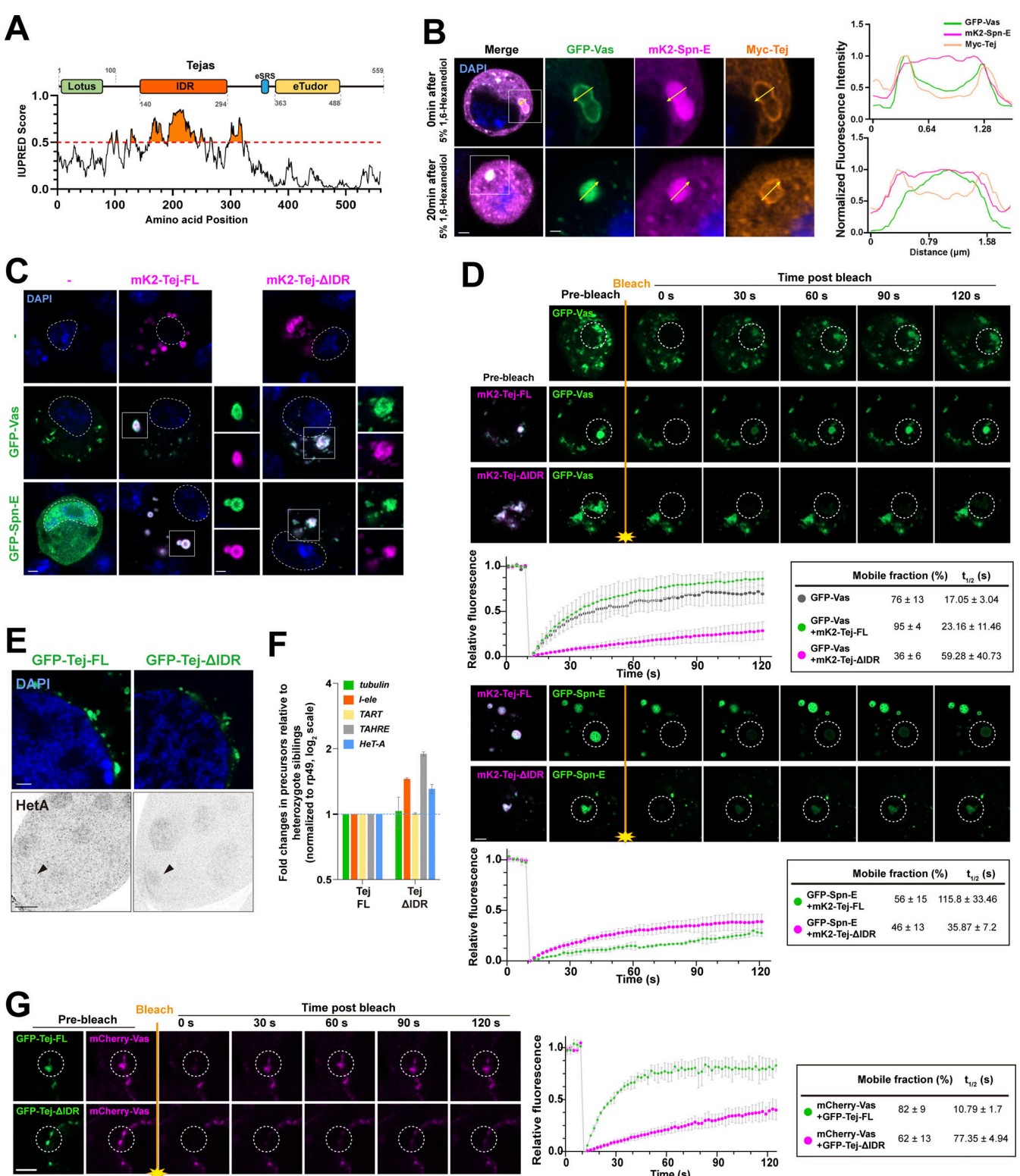

Figure 5. **Intrinsically disordered region (IDR) of Tej controls the mobility of nuage components. (A)** Schematic structure of Tej IDR using the IUPred prediction. The region with an IUPred score higher than 0.5 was defined as IDR. **(B)** Tej forms 1,6-hexanediol (1,6-HD)-sensitive core–shell granules with Vas and Spn-E. Myc-Tej is transfected with GFP-Vas and mK2-Spn-E in S2 cells. The fluorescence intensity of each protein is plotted in the diagram with or without 1,6-HD treatment. The intensity profiles of the designated lines (yellow arrows) are normalized to the highest value of each channel (right panels). **(C)** The recruitment of Vas and Spn-E to Tej condensates is independent of the Tej IDR domain. mK2-Tej-FL or Tej ΔIDR is transfected alone (magenta, top panels) or with GFP-Vas or Spn-E in S2 cells (green, middle, and bottom, respectively). The squares indicate aggregates of co-transfected proteins, and each protein is shown in the panels on the right. **(D)** The Tej IDR controls the mobility of the Tej aggregates observed with FRAP. mK2-Tej-FL or Tej ΔIDR (magenta) is transfected to S2 cells with GFP-Vas or Spn-E (green). The images show the recovery of the GFP signals for the aggregates before and after photobleaching

(dotted white circles). **(D and G)** The line graph shows the normalized relative recovery rate (bottom panel). The mean and ± SD are represented by colored dots and gray bars, respectively ($n$ = 3, number of analyzed independent experiments). The mean and ± SD value of proportion of the mobile fraction and $t_{1/2}$ derived from the fitting curves are shown in the table. **(E)** Immunostaining of *tej* mutant ovaries (*tej*[48–5]) expressing GFP-Tej-FL or Tej ΔIDR (top, green) for HeT-A Gap protein (bottom, arrowheads). **(F)** Fold changes of the transposon transcripts, *I-element*, *TART*, *TAHRE*, and *HeT-A*, and *tubulin* as the control, in *tej* mutant ovaries expressing Tej-FL or Tej ΔIDR. All values are normalized to *rp49*, and relative expression levels to those with Tej-FL are shown. Error bars indicate standard deviation ($n$ = 3, number of analyzed independent experiments). DNA is stained with DAPI (blue) in C and E. **(G)** Tej IDR facilitates the mobility of Vas in vivo, as observed by FRAP. GFP-Tej-FL or Tej ΔIDR (green) are overexpressed in the *tej* mutant ovary (*tej*[48–5]) with Vas-mCherry (magenta). The images show the recovery of the mCherry signals for the aggregates, before and after photobleaching (dotted white circles). Scale bars, 1 µm (B, right three panels), 1.5 µm (insets of panel C), 2 µm (B, left panels, C, and D), 20 µm (E), 5 µm (G).

Tej aggregation (Fig. 2 B and Fig. S6 A), which is consistent with previous studies showing that the eTudor domain is engaged in granulation by binding to its ligand sDMA (Courchaine et al., 2021).

Despite the unusual nuage granules of Tej-ΔeTudor, it mildly suppressed transposon expression (Fig. 4, B and F). Notably, Tej-ΔeTudor displays interaction with Vas and Spn-E, albeit to a lesser extent, especially with Spn-E (Fig. S5 C). Our CL-IP results also supported these interactions as reported in S2 cells (Patil and Kai, 2010). Alternatively, Tej-ΔeTudor possibly may facilitate the association of other components with nuage activity for piRNA processing. Unlike the mutation of precursor transporter, *nxf3* (ElMaghraby et al., 2019; Kneuss et al., 2019), and the ping-pong cycle assistant, *krimp* (Sato et al., 2015; Webster et al., 2015), *tej*, as well as *spn-E* and *vas* mutants, exhibited the accumulation of piRNA precursors in the perinuclear region and a collapse of the ping-pong amplification. These results suggest that they function upstream during ping-pong amplification. Stalling of piRNA precursors was also observed when the recruitment of Vas or Spn-E to nuage was abolished by the loss of the Lotus or eSRS domains, respectively. Precursor accumulation

was concentrated in the malfunctioning nuage or perinuclear region, which would result in a failure in precursor processing and cause TE upregulation.

Genetic analysis of nuage organization revealed that Spn-E and Tej occupy a higher hierarchical position than Vas at an earlier stage (Fig. S1 D), which is inconsistent with a previous observation (Patil and Kai, 2010), possibly due to the fluctuation of nuage assembly and/or structure at a later stage in the mutants. In contrast, Tej and Spn-E are mutually dependent for the proper assembly of nuage granules because Spn-E is required for the proper localization of Tej within nuage (Fig. S1 D and Fig. 4 B). Moreover, Tej may form a relatively stable scaffold with Spn-E for nuage assembly, while a mobile fraction of Tej may contain Vas. These results suggest that Tej may facilitate the compartmentalization of Vas and Spn-E, as shown in CL-IP experiments (Fig. 1 D) and also reported in *Bombyx* germ cells (Nishida et al., 2015), while we cannot exclude the possibility of simultaneous binding among these proteins. Our further results with S2 revealed that the weak hydrophobic interaction between the proteins may contribute to the formation and regulation of membraneless structures on nuage. DEAD-box RNA helicase

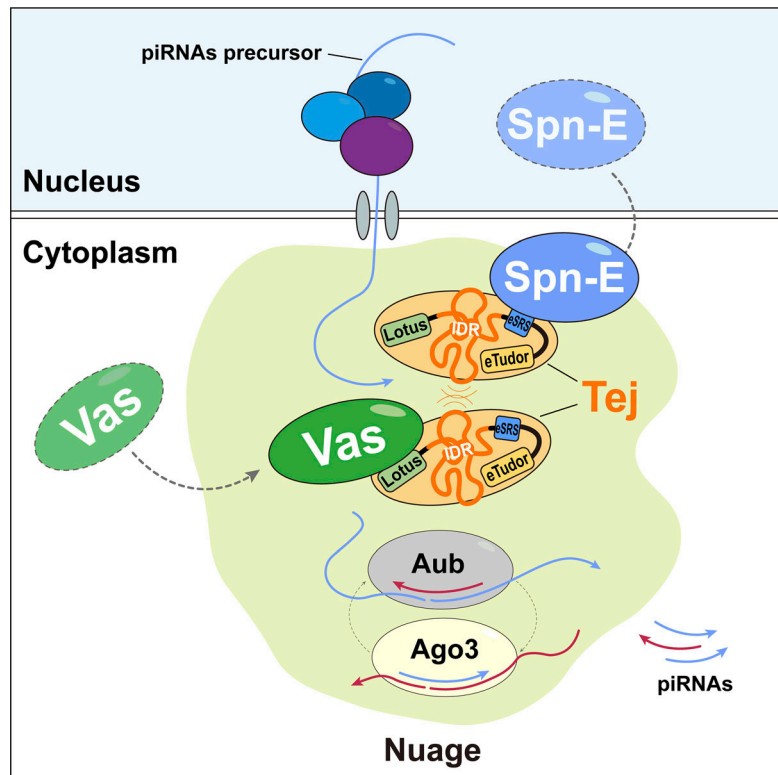

Figure 6. **A model: The Tej-mediated nuage organization is essential for the piRNA processing pathway.** In the *Drosophila* germ cells, intrinsically nuclear-localized Spn-E is recruited or intercepted by Tej via its eSRS motif and settles together on the cytoplasmic side of the perinuclear region for further nuage organization. Tej aggregates to form nuage granules, depending on the physical interaction or IDR-mediated hydrophobic interaction. Thereafter, through the Lotus domain of Tej, Vas is further recruited and employed for piRNA precursor processing in the nuage. Tej IDR modulates the dynamics of other nuage components, especially Vas, in membraneless nuage for efficient piRNA production.

family members, including Vas homolog, reportedly form non-membranous, phase-separated organelles in both prokaryotes and eukaryotes (Hondele et al., 2019), and the large IDR at the N-terminal region facilitates their aggregation by LLPS (Nott et al., 2015). In addition, the loss of IDR in Tej significantly suppressed the mobility of Tej and Vas; nevertheless, the TE repression was only mildly attenuated (Fig. 5 F). Thus, Tej-ΔIDR may remain colocalized with Vas and Spn-E, facilitating the processing of piRNAs (Fig. 5 C). Alternatively, the reduction of Vas mobility by the loss of Tej IDR could be compensated by other components in nuage. Only the localization of Vas was remarkably changed upon 1,6-HD treatment in S2 cells, further supporting the finding that weak hydrophobic interaction controlled the dynamics of Vas, although we cannot exclude a possibility of the unexpected effects by the 1,6-HD treatment. We also cannot exclude the possibility that 1,6-HD treatment might have impaired kinase and/or phosphatase activity (Düster et al., 2021). Hence, localization might have been affected by the changes in their phosphorylation status. The behavior of these proteins is seemingly influenced by their respective binding modes and properties with Tej. The interaction of Vas with Tej is affected by 1,6-HD and IDR region of Tej through the hydrophobic association, whereas that of Spn-E with Tej is more rigid, possibly contributing to the formation of the scaffold of nuage. In conclusion, Tej utilizes the eTudor domain for granule formation, whereas the IDR of Tej appears to maintain the assemble of Tej granules, controlling the mobility of Vas in nuage.

Membraneless macromolecular nuage contains more than a dozen components, including Vas and Tej that harbor IDRs, which could contribute to the dynamics of nuage and impact the efficient production of piRNAs. Nuage also contains piRNA precursors and TE RNAs that are processed therein; their unique or specific propensities may affect nuage assembly and function. Further investigation of those proteins and RNA components will shed light on the regulatory mechanisms underlying the formation and dynamics of nuage to promote each sequential step of piRNA biogenesis.

# Materials and methods

## Fly stocks
All stocks were maintained at 25°C with standard methods. The mutant alleles used in the study were *tej⁴⁸⁻⁵* (Patil and Kai, 2010), *vas^PH165* (Styhler et al., 1998), *spn-E⁶¹⁶* (Ott et al., 2014), *krimp^f06583* (BL #18990; Lim and Kai, 2007), *ago3^t2/t3* (BL #28269; BL #28270; Li et al., 2009), *aub^QC42/HN2* (BL #4968; BL #8517; Schüpbach and Wieschaus, 1991), *nxf3^Δ* (BL #90328; Kneuss et al., 2019), *Df(2L) BSC299* (BDSC #23683), and *Df(3R)Exel8162* (BL #7981). Driver lines for germline and somatic gonadal cells were *NGT40*-Gal4; *nos*-Gal4 VP16 (Grieder et al., 2000), and *Traffic jam*-Gal4 (DGRC #104055; Hayashi et al., 2002), respectively. Either *yw* or the respective heterozygote was used as a control. Knock-In fly lines *vas^mCherry.HA.KI* (DGRC #118618), *vas^EGFP.KI* (DGRC #118616), and *aub^EGFP.KI* (DGRC #118621; Kina et al., 2019) were obtained from the Drosophila Genetic Resource Center at the Kyoto Institute of Technology, Japan. All *Drosophila* genotypes used in this study are listed in Table S1.

## Generation of knock-in fly lines
Tej-GFP, mKate2-Ago3, and Spn-E-mKate2 knock-in fly lines were generated through CRISPR-Cas9-induced double-strand breaks restored by the homology-directed repair (HDR) in the presence of donor plasmids. Two guide RNAs were designed to direct the Cas9 proteins to the regions flanking the start/stop codon of each target gene to induce a big scale of double-strand breaks. The following guide RNA sequences were cloned into pDCC6 (Gokcezade et al., 2014): Tej-GFP gRNA1, 5′-GATCGCTCATAGAAACTGGT-3′; Tej-GFP gRNA2, 5′-GTGCATAGATTTCTATTATA-3′; mK2-Ago3 gRNA1, 5′-TAATAAAAATGCTGGCAATA-3′; mK2-Ago3 gRNA2, 5′-TGTGTGTTTCAGAGCATGTC-3′; Spn-E-mK2 gRNA1, 5′-GATCACGATGCAATATGGTC-3′; Spn-E-mK2 gRNA2, 5′-GAACGATGTAACCATTCTTAT-3′. Donor vectors containing the GFP or mKate2 coding sequence flanked by 1-kb homology arms adopted from both 3′ and 5′ sides of the insertion site were generated by cloning of PCR-amplified tags and arms into linearized pGEM-3z vector by In-Fusion HD Cloning Kit (Takara Bio). Obtained gRNA expression plasmids and donor plasmids were injected into the *y w* embryos with a final concentration of 120 ng/μl for each. The knock-in events positive founders and progenies were confirmed by single fly genome PCR genotyping. Tej-GFP, mKate2-Ago3, and Spn-E-mKate2 Knock-In flies were crossed with the corresponding loss-of-function allele *tej⁴⁸⁻⁵*, *Ago3^t2*, and *Spn-E⁶¹⁶* for checking the functionality of endogenies fusion proteins, and all the fluorescence-fused proteins rescued their corresponding loss-of-function alleles. Homozygous fly lines containing *tej^EGFP.KI*, *spn-E^mKate2.KI*, and *ago3^mKate2.KI* were viable and fertile, suggesting a negligible impact of fluorescent tag on their functions.

## Generation of transgenic fly lines
The transgenic fly lines containing miniTurbo-GFP-tagged Tej-FL, Tej-ΔLotus, Tej-ΔeTudor, Tej-ΔeSRS, and GFP-tagged Tej-ΔSRS, Tej-ΔIDR, Spn-E-FL, Spn-E-ΔNLS were generated by PhiC31 integrase-mediated transgenesis system. The constructs for injection were generated using the cDNAs obtained by reverse transcription from ovarian RNA of *y w* flies. DNA fragments of GFP and the respective variants were amplified and cloned into the pUAS-K10-*attB* plasmid backbone (Koch et al., 2009). The transgenic constructs were injected into the embryo of *attP*-containing strains (P40, BDSC #25709 and P2, BDSC #25710), and progenies expressing mini white were obtained. For rescue experiments, transgenes were recombined with *tej⁴⁸⁻⁵* or *Spn-E⁶¹⁶*/Df background and driven by the germline driver *NGT40*-Gal4; *nos*-Gal4-VP16, or the ovarian somatic cell driver, *traffic jam*-Gal4.

## Antibody generation
Rat anti-Spn-E, Rabbit anti-HeT-A-Gag, rat anti-Ago3, and rat anti-Tej were generated in this study. N-terminal GST-tagged Spn-E (4–450th aa) antigen peptide was expressed in *Escherichia coli* strain BL21 (DE3) by IPTG, with the plasmid generously provided by Dr. M. Siomi (University of Tokyo, Tokyo, Japan). The GST-Spn-E antigen peptide purified by GST affinity beads was used to immunize rats. The Spn-E antibody was further purified from the rat sera with the GST affinity beads conjugated

with GST-Spn-E antigen peptide and stocked in 50% (vol/vol) glycerol at –20°C. The plasmid, including the fragment that encodes a part of HeT-A-gag (201 amino acids), was generously provided by Dr. Mary-Lou Pardue (Massachusetts Institute of Technology, Cambridge, MA, USA). DNA fragments encoding the N terminal of Tej (1–110th aa) and Ago3 (1–150th aa) were amplified from the cDNA, cloned into pENTR/D-TOPO plasmids, and recombined into either pDEST15 or pDEST17 (Invitrogen). Primers used for the cloning are as follows: HeT-A gag fw; 5′-CACCCCCTACTGGAAAAGCTGAAC-3′, HeT-A gag rv; 5′-CTA CAGGGCATCCTTTGTACGCGCT-3′, Tej antigen fw; 5′-ATGGAT GATGGAGGGGAGTT-3′, Tej antigen rv; 5′-CTCGGAGGCGTA GCAATA-3′, Ago3 antigen fw: 5′-ATGTCTGGAAGAGGAAA-3′, Ago3 antigen rv; 5′-TTACACTTCGTAATTAAAAA-3′. The antigens were expressed in *E. coli* strain BL21 (DE3) by IPTG. The purified soluble His-HeT-A-Gag and GST-Tej antigen peptides and the gel pieces of the insoluble GST-Ago3 antigen peptide from SDS-PAGE gel were used to immunize animals (Eve Bioscience). Rabbit serum against HeT-A-gag peptide was directly used for immunostaining. The Tej and Ago3 antibodies were further purified from the sera; insoluble His-Tej antigen and the GST-Ago3 antigen peptide-containing region blotted on PVDF membrane (WAKO) were sliced into pieces and incubated with the sera at 4°C overnight with rotation. After incubation, the membrane pieces were washed in 1% (vol/vol) PBS-Tween for 2 h at room temperature and the antibodies were eluted with 0.1 M glycine-HCl (pH 2.5). The elutes were neutralized to pH 7.0 by NaOH and stocked in 50% (vol/vol) glycerol at –20°C.

## Western blotting

The ovaries were homogenized in the lysis buffer containing 30 mM HEPES (pH 7.4), 80 mM KOAc, 2 mM DTT, 10% (vol/vol) glycerol, 2 mM MgCl₂, and 0.1% (vol/vol) Triton X-100. After centrifugation at 20,600 ×*g* for 10 min at 4°C, the supernatants were electrophoresed through pre-cast 5–20% e-PAGEL gels (ATTO) and transferred to ClearTrans SP PVDF membrane (Wako). The primary and secondary antibodies used in this study are listed in Table S2. Antibodies were diluted and stored in the Signal Enhancer reagent HIKARI (NACALAI TESQUE). Chemiluminescence was induced by the Chemi-Lumi One reagent kit (NACALAI TESQUE), and immunoreactive bands were detected using ChemiDoc Touch (Bio-Rad Laboratories) and processed by ImageJ (Fiji).

## Small RNA immunoprecipitation

For IP of Aub- and mK2-Ago3-bound-piRNAs, 200 ovaries were dissected manually from adult flies in chilled PBS and homogenized with the lysis buffer containing 20 mM Tris-HCl (pH 7.4), 200 mM NaCl, 2 mM DTT, 10% (vol/vol) glycerol, 2 mM MgCl₂, 1% (vol/vol) Triton X-100, 1× cOmplete protease inhibitor cocktail (Roche), and 1% (vol/vol) RNaseOUT recombinant ribonuclease inhibitor (Invitrogen). The lysates were cleared by centrifugation at 20,600 ×*g* for 10 min at 4°C three times to remove the contamination of the lipid. Mouse anti-Aub antibody (1:20; Patil and Kai, 2010) or mouse anti-mKate2 (1:200; Evrogen) was added to the cleared lysate and incubated at 4°C for 2 h with rotation. Then Dynabeads Protein G/A (Invitrogen) was

added to the lysate-antibody mixture and incubated at 4°C for 1 h with rotation. After incubation, the magnet beads were collected and washed at least four times with a washing buffer containing 20 mM Tris-HCl (pH7.4), 400 mM NaCl, 2 mM DTT, 10% (vol/vol) glycerol, 2 mM Mgcl₂, 1% (vol/vol) Triton X-100, 1× cOmplete protease inhibitor cocktail (Roche), and 1% (vol/vol) RNaseOUT recombinant ribonuclease inhibitor (Invitrogen). 10% of the precipitates were analyzed by Western blotting to check the protein immunoprecipitation efficiency. RNAs were isolated from the rest 90% of the precipitates with TRIzol LS (Invitrogen) according to the standard manufacturer's protocol. Purified small RNAs were labeled with ³²P-γ-ATP using T4 polynucleotide kinase (Thermo Fisher Scientific). After electrophoretic separation by 15% urea-containing denaturing polyacrylamide gel in ×0.5 TBE, radioisotope signals were captured and analyzed by Amersham Typhoon scanner (GE), further processed by ImageJ (Fiji).

## Analysis of small RNA libraries

Small RNA libraries were sequenced using Illumina HiSeq-2500 according to the manufacturer's protocol at Genome Information Research Center, Research Institute for Microbial Diseases of Osaka University. Small RNA reads were normalized with noncoding RNAs including snoRNAs, snRNAs, miRNAs, and tRNAs. After trimming (5′ adaptor: 5′-AGATCGGAAGAGCAC ACGTCT-3′) and removing rRNA, snoRNAs, snRNAs, miRNAs, and tRNAs, 23- to 29- nt reads were mapped to the piRNA clusters or transposable elements with up to 3-nt mismatching by Bowtie (Langmead et al., 2009). piRNA cluster definition was referred to those previously reported (Brennecke et al., 2007), and TE sequences were adopted from the Flybase (Release 6.32). The normalized numbers of cluster-mapping reads were distributed to the position of the cluster sequence and visualized with pyGenomeTracks (Ramírez et al., 2018). The sequence logos were generated by using ggplot2 R package ggseqlogo (Wagih, 2017).

## Crosslinking immunoprecipitation

Ovaries were manually dissected in ice-chilled PBS, fixed with PBS containing 0.1% (wt/vol) paraformaldehyde for 20 min on ice, quenched in 125 mM glycine for 20 min, and then homogenized in crosslinking immunoprecipitation (CL-IP) lysis buffer containing 50 mM Tris-HCl (pH 8.5), 150 mM KCl, 5 mM EDTA, 1% (vol/vol) Triton X-100, 0.1% (wt/vol) SDS, 0.5 mM DTT, and 1× cOmplete protease inhibitor cocktail (Roche). The lysate was incubated at 4°C for 20 min with rotation, followed by 30 s sonication with a Bioruptor three times with 30-s intervals for cooling (Sonicbio). After centrifugation at 20,600 ×*g* for 10 min at 4°C, the supernatant was collected in new Eppendorf Protein LoBind tubes and diluted with equal volumes of CL-IP wash buffer containing 25 mM Tris-HCl (pH 7.5), 150 mM KCl, 5 mM EDTA, 0.5% (vol/vol) Triton X-100, 0.5 mM DTT, and 1× cOmplete protease inhibitor cocktail (Roche). The diluted lysate was precleaned by Dynabeads Protein G/A (Invitrogen) 1:1 mixture for 1 h at 4°C and incubated with the antibody (mouse anti-GFP [3E6, 1:500; Thermo Fisher Scientific] or mouse anti-mKate2 [AB233, 1:500; Evrogen]) overnight at 4°C. Dynabeads Protein

G/A (Invitrogen) equilibrated with CL-IP washing buffer (1:1 mixture) was added to the lysate–antibody mixture, incubated at 4°C for 3 h with rotation, collected, and washed at least four times with the CL-IP washing buffer. When required harsh binding and washing conditions, the potassium salt concentration of the CL-IP washing buffer was adjusted up to 1 M. After washing, bead-bound proteins were retrieved by suspending with the equal volume of the SDS containing 2× sample buffer, heated at 95°C for 5 min, and analyzed through 12% SDS-PAGE gels for Western blotting. Chemiluminescence was induced by the Chemi-Lumi One reagent kit (NACALAI TESQUE). Immunoreactive bands were detected by ChemiDoc Touch (Bio-Rad Laboratories), and processed and quantified by ImageJ (Fiji).

### RT-qPCR
Total RNAs were extracted from the 2-d-old ovaries fattened up with yeast paste with TRIzol LS (Invitrogen) according to the manufacturer's protocol and treated with DNase I (Invitrogen). cDNAs were generated by reverse transcription with Super-Script III system (Invitrogen) using oligo d(T)20 and hexadeoxyribonucleotide mixture primer. qPCR was performed using KAPA SYBR Fast qPCR Master Mix (KAPA biosystems). All the expression levels of examined genes were normalized to that of $rp49$. The primer sequences for detecting transposon transcripts and piRNA cluster transcripts are shown in Table S3.

### S2 cell culture experiments
*Drosophila* Schneider S2 cells were grown at 26°C in 10% (vol/vol) fetal bovine serum (FBS)-supplemented Schneider medium, with the presence of 50–100 U penicillin and 50–100 µg streptomycin. Plasmids used for transfection were generated using the Gateway cloning system (Life Technologies): transgenes were recombined with the *Drosophila* Gateway Vector Collection (DGVC) destination vectors expressing the N-terminal tag fused target proteins under Actin5C promoter (Invitrogen). In addition, a new destination vector for the expression of mKate2-tagged protein at the N-terminus under Actin5C promoter, pAKW, was constructed in this study. Transfected S2 cells were placed onto the concanavalin A precoated coverslips, incubated at 26°C for at least 20 min for an efficient adhesion, fixed for 15 min in 4% (wt/vol) paraformaldehyde, permeabilized for 10 min in PBX (PBS with 0.2% [vol/vol] TritonX-100), and washed for 10 min by PBX twice. DNA was stained with DAPI (1:1,000) for 10 min and rinsed with PBS and equilibrated in Fluoro-KEEPER Antifade Reagent (NACALAI TESQUE) for 10 min before mounting. Images were taken by ZEISS LSM 900 with Airy Scan 2 using 63× oil NA 1.4 objectives and processed by ZEISS ZEN 3.0 and ImageJ (Fiji).

### Immunofluorescence staining
Immunostaining of ovaries was conducted as previously reported (Lim et al., 2022). The antibodies used for immunostaining are listed in Table S2. Secondary antibodies were Alexa Fluor 488-, 555-conjugated goat anti-rabbit and anti-mouse IgG (A11034, A21428, A21127; Thermo Fisher Scientific), 1:200 diluted in 0.4% (wt/vol) BSA containing PBX as the working solution. Ovaries expressing endogenous fluorescent-tagged proteins were fixed with PBS containing 0.1% (wt/vol)

paraformaldehyde for 20 min on ice and further washed with PBX (PBS with 0.2% [vol/vol] Triton X-100) for 10 min twice. DNA were stained with DAPI (1:1,000) for 10 min, rinsed with PBS, and equilibrated in Fluoro-KEEPER Antifade Reagent (NACALAI TESQUE) for 10 min before mounting. Images were taken by ZEISS LSM 900 with Airy Scan 2 using 63× oil NA 1.4 objectives and processed by ZEISS ZEN 3.0 and ImageJ (Fiji).

### RNA in situ hybridization chain reaction (HCR)
The probes targeting the transcripts derived from the unique regions at *cluster 38C* (Chr2L: 20104896..20213637) and *42AB* (Chr2R: 6322410..6323756) and the reagents were purchased from Molecular Instruments, Inc. The protocol was modified from what was previously reported (Slaidina et al., 2020). Ovaries were fixed in 4% formaldehyde for 20 min, washed twice with PBST at room temperature, and dehydrated by sequential washing with 25, 50, 75, and 100% (vol/vol) methanol in PBS for 5 min each on ice. Dehydrated ovaries were stored at –20°C overnight and rehydrated by sequential washes with 100%, 75%, 50%, and 25% (vol/vol) methanol in PBS on ice. Samples were prewarmed for 2 h in PBX at room temperature, followed by post-fixation with 4% (wt/vol) paraformaldehyde, and sequentially washed as follows: twice with PBST for 5 min on ice, once with 50% (vol/vol) PBST and (vol/vol) 50% 5× SSCT (5× SSC with 0.1% [vol/vol] Tween-20) for 5 min on ice, and twice with 5× SSCT for 5 min on ice. Then, the ovaries were equilibrated with the hybridization buffer for 5 min on ice, prehybridized in the hybridization buffer for 30 min at 37°C, and incubated with 0.5 ml of prewarmed probe hybridization buffer containing 4 pmol of the probes overnight in a light-avoiding 37°C shaker. After hybridization, ovaries were washed four times with the probe washing buffer for 15 min each at 37°C and twice with 5× SSCT for 5 min each at room temperature. Next, the ovaries were equilibrated in a prewarmed amplification buffer for 5 min at room temperature. 30 pmol of the probes were denatured at 95°C for 90 s and chilled down to room temperature for 30 min. Then the hairpins were cooled on ice for 10 s and mixed with 500 µl amplification buffer at room temperature. The chain reaction was conducted by incubating the ovaries in a freshly prepared probe solution overnight in a light-avoiding container at room temperature and terminated by washing twice with 5× SSCT for 5 min. Then the samples were washed in 5× SSCT containing DAPI (1:1,000) and Alexa Fluor 488-conjugated Wheat Germ Agglutinin (WGA, 5 µg/ml; Thermo Fisher Scientific) and with 5× SSCT for 30 min each at room temperature. The ovaries were equilibrated in Fluoro-KEEPER Antifade Reagent (NACALAI TESQUE) at room temperature before mounting (Choi et al., 2018; Slaidina et al., 2020). Images were taken by ZEISS LSM 900 with Airy Scan 2 using 63× oil NA 1.4 objectives and processed by ZEISS ZEN 3.0 and ImageJ (Fiji).

### Quantification analysis of in situ-HCR signal for piRNA precursors
Each image was processed and quantified with ImageJ (Fiji). Fluorescence intensity of cluster *38C* or *42AB* transcripts by HCR-FISH was measured and quantified after background subtraction. The annular region of ±5% nuclei diameter inside and

outside of the nuclear membrane stained by WGA was defined as the perinuclear region.

### Fluorescence recovery after photobleaching (FRAP)

Transfected S2 cells were placed in a concanavalin A-precoated multi-well glass-bottom culture chamber (MATSUNAMI) for over 30 min at 26°C. Ovaries were dissected in prewarmed 10% (vol/vol) fetal bovine serum (FBS)-supplemented Schneider medium. The muscle sheath was removed from ovarioles and they were distributed in the prewarmed medium with 10 mg/ml fibrinogen (Millipore) and placed in a glass-bottom dish (MATSUNAMI). Add 1 μl thrombin (10 U/ml; GE Healthcare Lifesciences) to the medium drop for forming the fibrinogen–thrombin clot which fixes the ovarioles (Wilcockson and Ashe, 2021). All images were taken at 26°C in the incubation modules advanced ZEISS LSM 900 with Airy Scan 2 using 63× oil NA 1.4 objectives and processed by ZEISS ZEN 3.0 and ImageJ (Fiji). One single granule that has GFP signals in each cell was repeatedly bleached using a pulse of 488 or 561 nm lasers 50 times within 3 s, and images were taken every second to record fluorescence intensity. Initial 10 images were acquired to establish the levels of prebleach fluorescence. Fluorescent intensity by bleaching in the specific ROI was analyzed with easyFRAP (Rapsomaniki et al., 2012). A full-scale normalization procedure was used to correct differences in bleaching depth among different experiments and the recovery curves. Individual normalized data were fitted with a double-term exponential equation and used for the calculation of the half-time of full fluorescence recovery ($t_{1/2}$[s]) and the percentage of mobile fraction. Each value is averaged and represented in the table (Rapsomaniki et al., 2012).

### Protein disorder prediction and conservation analysis

The intrinsically disordered region was analyzed with the IUPred server (https://iupred2a.elte.hu/). The region containing residues with IUPred scores more than 0.5 was classified as a prominent intrinsically disordered region (Dosztányi et al., 2005; Mészáros et al., 2018).

### Online supplemental material

This manuscript is accompanied by six supplementary figures. Fig. S1 contains data on the validation of endogenously tagged nuage components for Fig. 1. Fig. S2 contains data supporting Fig. 2. It shows the identification of the interacting domain of Tej to Spn-E and the predicted structure by AlphaFold v2.2. Fig. S3 contains data supporting Fig. 2. It shows five different predictions of interacting Tej and Spn-E by AlphaFold v2.2 and their PAE plots. Fig. S4 contains data supporting Fig. 3. It contains Aub- and Ago3-bound piRNAs in *tej* mutant ovaries. Fig. S5 contains data supporting Fig. 4. It shows ovaries expressing Tej variants and the analysis of their interactions with Spn-E and Vas. Fig. S6 contains data supporting Fig. 5. It shows ovaries expressing Tej variants and FRAP analysis of each variant in S2 cells and Vas and Spn-E in ovaries. Table S1 shows *Drosophila* genotypes used in this study. Table S2 is a list of antibodies used in this study. Table S3 is a list of primers used for qRT-PCR in this study.

### Data availability

The data underlying Fig. 3 and Fig. S3 are openly available in DNA Data Bank of Japan (DDBJ), BioProject Accession: PRJDB13876/DRA Accession: DRA016848. The other data are available in the published article and the online supplemental material. All fly strains and antibodies generated for this study are available upon request.

## Acknowledgments

We are grateful to Dr. Mandy Jeske (BZH, University Heidelberg, Heidelberg, Germany) for useful discussion about unpublished data and comments for the manuscript, to Dr. Mikiko C. Siomi (University of Tokyo, Tokyo, Japan) for generous gifts of plasmid that encode Spn-E antigen, to Dr. Mary-Lou Pardue (Massachusetts Institute of Technology, Cambridge, MA, USA) for providing plasmid encoding Het-A-gag, to Isshiki Wakana for assisting Spn-E knock-in fly generation. We acknowledge Bloomington Drosophila Stock Centre and Kyoto Stock Center for the fly stocks. We also thank the Cybermedia Center, Osaka University for performing the SpnE-Tej structure prediction by using large-scale computer systems. We appreciate the insightful discussion and suggestions from all the members of the Kai laboratory.

This work was supported by Grant-in-Aid for Scientific Research B (21H02401) for T. Kai, TAKEDA Bioscience Research Grant (J191503009) for T. Kai, Osaka University Institute for Datability Science "Transdisciplinary Research Project" (Na22990007) for S. Kawaguchi and T. Kai, and Grant-in-Aid for Transformative Research Areas (A) (21H05275) for T. Kai.

Author contributions: Conceptualization: Y. Lin, R. Suyama, and T. Kai; Data Curation: Y. Lin and R. Suyama; Formal Analysis: Y. Lin, R. Suyama, and S. Kawaguchi; Funding Acquisition: S. Kawaguchi and T. Kai; Investigation: Y. Lin; Methodology: Y. Lin, R. Suyama, S. Kawaguchi, T. Iki, and T. Kai; Project Administration: T. Kai; Resources: T. Iki, T. Kai; Software: Y. Lin, R. Suyama, and S. Kawaguchi; Supervision: R. Suyama and T. Kai; Validation; Y. Lin, R. Suyama, T. Iki, and T. Kai; Visualization: Y. Lin, R. Suyama, and S. Kawaguchi; Writing, Original Draft: Y. Lin and R. Suyama; Writing, Review and Editing: Y. Lin, R. Suyama, and T. Kai.

Disclosures: The authors declare no competing interests exist.

Submitted: 27 March 2023

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

# Supplemental material

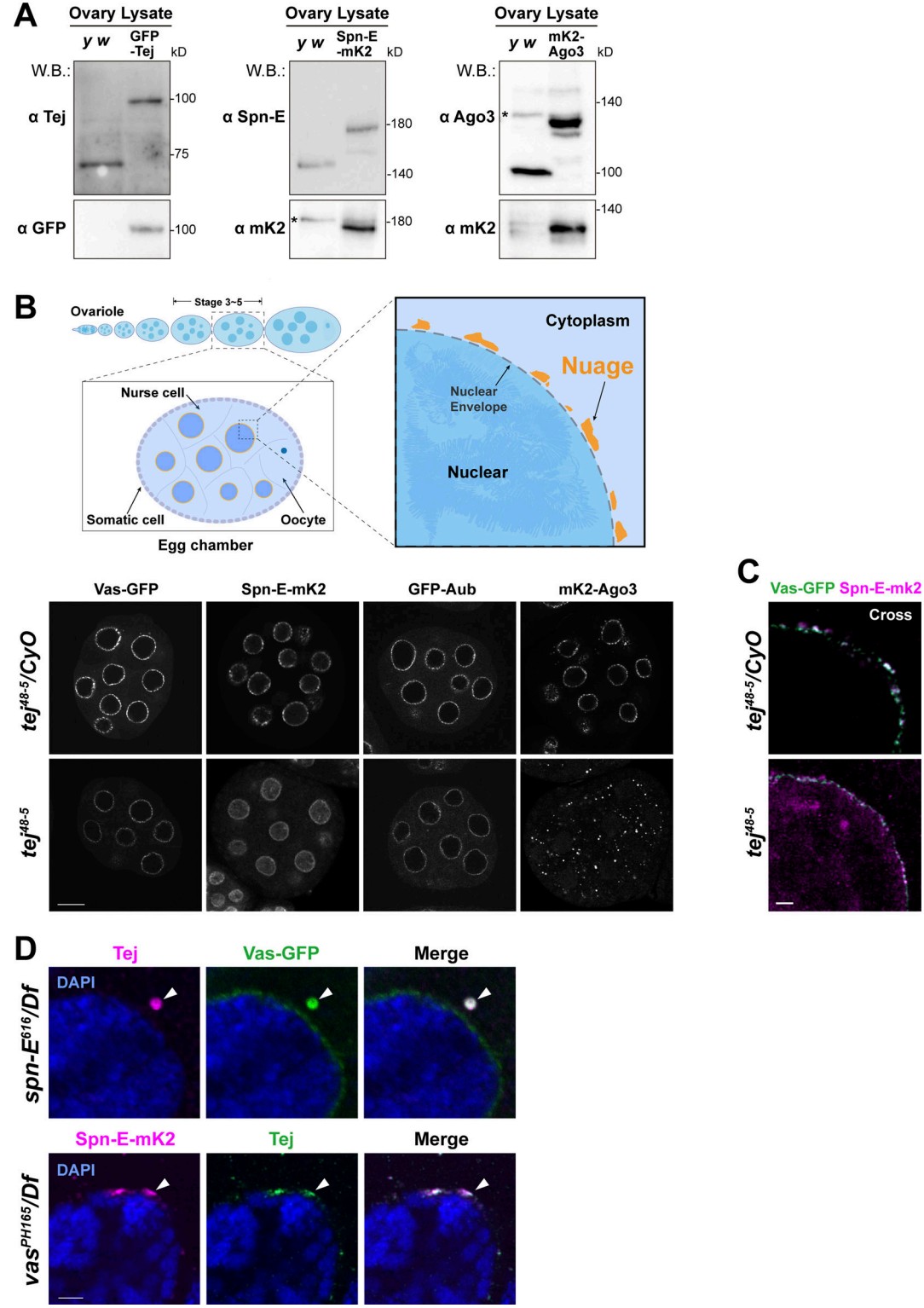

Figure S1.  **Endogenously tagged nuage components are expressed properly in the ovaries. (A)** Western blot analysis of ovarian lysates showing the expression of the newly generated GFP-Tej, Spn-E-mK2, and mK2-Ago3 using genome editing, as detected with either individual antibodies or fluorophore proteins. Ovaries of *y w* flies are used as a control. Asterisks denote unspecific bands. **(B)** Schematic drawings (upper panels) and confocal images (lower panels) of ovarioles, stage 5 egg chambers, and magnified germline cell showing perinuclear nuage of *Drosophila*. Fluorescent-tagged nuage components are present but their localization is affected in *tej* mutant ovaries (*tej*^48–5^, bottom panels). All the immunofluorescence signals are represented in grayscale. **(C)** A cross-section of the nurse cell nucleus of each genotype. Colocalization of Vas-GFP and Spn-E-mK2 is lost when Tej is absent (*tej*^48–5^). Scale bars, 10 µm (B), 1 µm (C). **(D)** Immunostaining of the ovaries expressing Spn-E-mk2 (magenta) in *vas* mutant (*vas*^PH165^/*Df*) and Vas-GFP (green) in *spn-E* mutant (*spn-E*^616^/*Df*), respectively, for Tej. The DNA is stained with DAPI (blue). The arrowheads denote Tej granules containing Vas or Spn-E. Scale bars, 1 µm (A, left and right top two images), 10 µm (A, bottom right image), 1 µm (D).

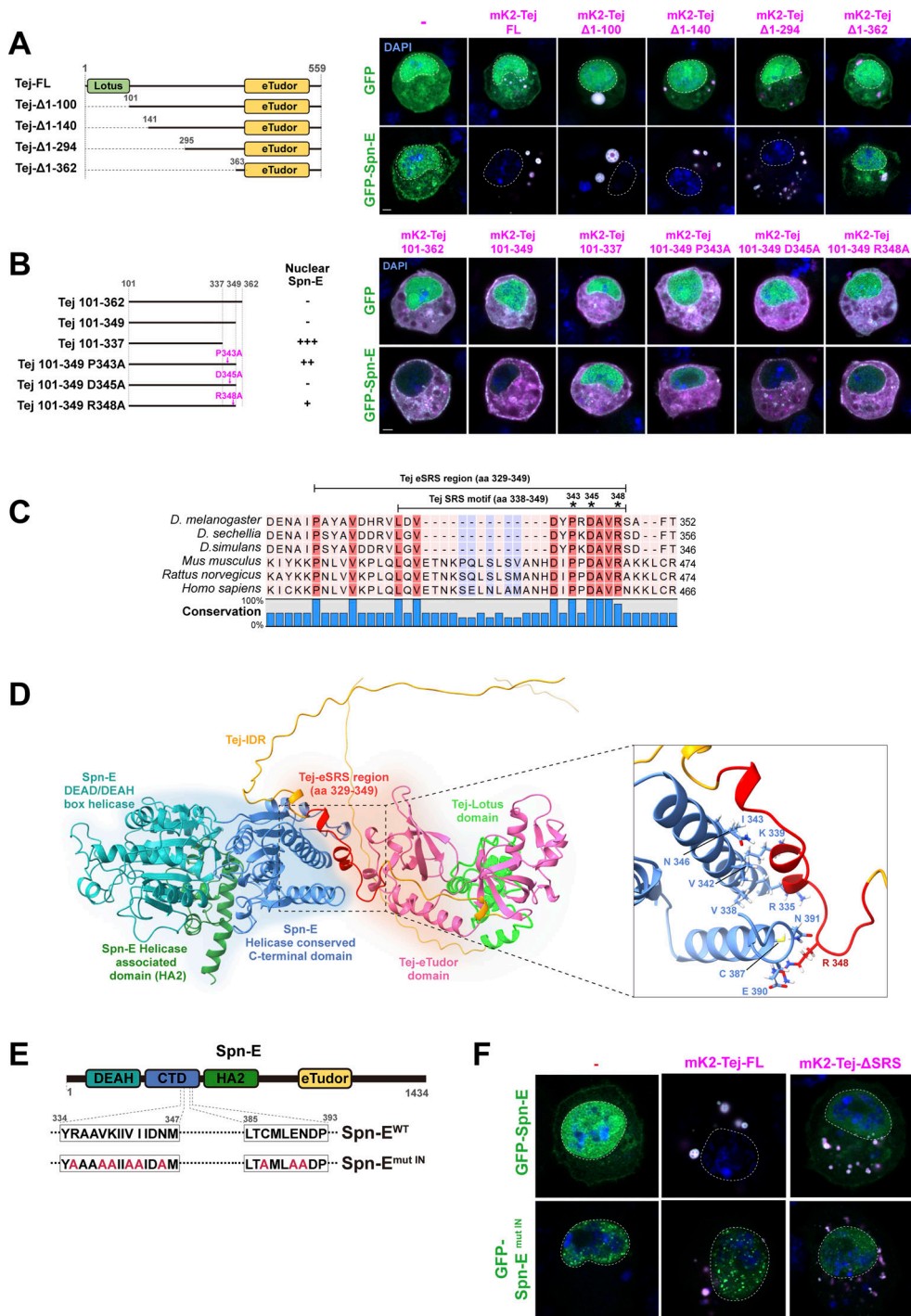

Figure S2. **Subcellular localization of Spn-E is controlled by Spn-E NLS and Tej. (A)** Identification of the Spn-E-recruiting domain in Tej. Tej truncated variants are in the schematic representation on the left: full-length Tej and Tej devoid of the first 100, 140, 294, and 362 aa are denoted as FL, Δ1–100, Δ1–140, Δ1–294, and Δ1–362, respectively. GFP or GFP-Spn-E (green, top, and bottom panel, respectively) is co-expressed with mK2-Tej-FL or its truncated variants (magenta) in S2 cells. Single transfections of these are shown in the panels on the left. **(B)** Truncation analysis to search for the regions associated with Spn-E. Schematic representation of Tej middle part variants (left); Proline, aspartic acid, and arginine at positions 343, 345, and 348 are mutated to alanine, respectively. Either GFP alone or GFP-Spn-E (green) is co-expressed with the truncated variants of mK2-tagged Tej in the middle part (magenta). **(C)** Schematic representation of the highly conserved amino acids in Tej. A bar graph of the conservation ratio in percentiles is shown at the bottom. The SRS and eSRS regions are indicated by lines, and asterisks mark single amino acid substitutions. **(D)** The part of the Rank 0 model (in S3A) is shown with more details. The inset represents an enlarged view of the interface of Tej and Spn-E interaction. The numbered residues in Spn-E (blue) that are predicted to interact with eSRS (red) are mutated in the subsequent experiments. **(E)** The schematic representation of Spn-E(mut IN)-containing substituted residues (red) that were predicted to be on the interface with Tej SRS. **(F)** Mutations at the Spn-E interface and deletion of the Tej SRS abolish the recruitment of Spn-E to the cytoplasm. GFP-Spn-E or Spn-E(mut IN) (green) is co-expressed with mK2-Tej-FL or Tej-ΔSRS (magenta) in S2 cells. The DNA is stained with DAPI (blue) in A, B, and F. Scale bars, 2 μm (A, B, and F).

**A**

Rank 0
Rank 1
Rank 2
Rank 3
Rank 4

Rank 0="model_3_multimer_v2_pred_0": 0.6429424908619761
Rank 1="model_1_multimer_v2_pred_0": 0.6361931429853533
Rank 2="model_4_multimer_v2_pred_0": 0.5944704494978156
Rank 3="model_5_multimer_v2_pred_0": 0.5943308167499957
Rank 4="model_2_multimer_v2_pred_0": 0.5717023253643091

**B**

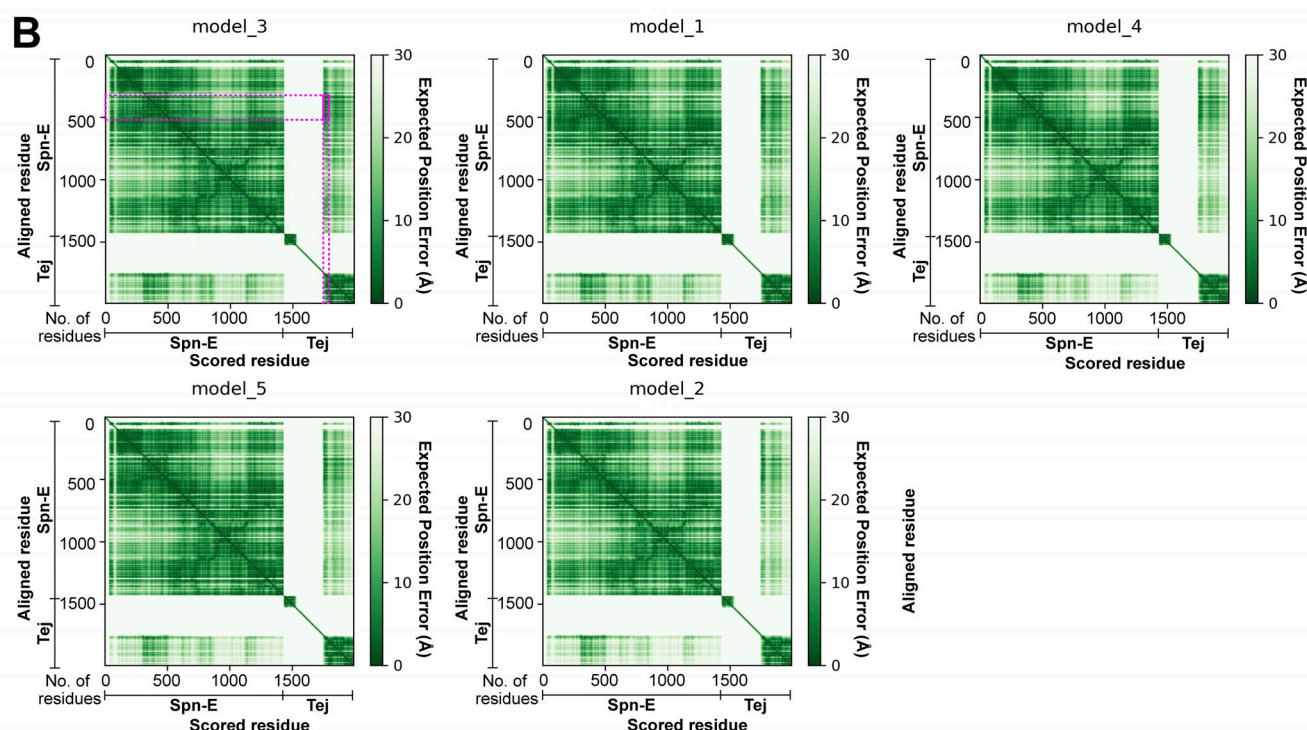

Figure S3. **Predicted structures for Tej by AlphaFold v2.2. (A)** The predicted structures of the heterodimer between Spn-E (left, in gray) and Tej (right, in various colors) using AlphaFold v2.2. Five predicted models superimposed with Spn-E and Tej structures are colored by Ranking. **(B)** The corresponding PAE plots of the predicted structures in A are shown. The eSRS of Tej aligned to the predicted interface on Spn-E is marked by a magenta frame in model 3 (Rank 0) showing lower error scores.

Figure S4.  **Aub- and Ago3-bound piRNAs are remarkably reduced in *tej* mutant ovaries. (A and B)** The piRNAs extracted from the immunoprecipitated Aub (A) and mK2-Ago3 (B) in the control (*tej*[48–5]/CyO) and *tej* mutant ovaries (*tej*[48–5]) are visualized via [32]P-labeling. The immunoprecipitated Aub or mK2-Ago3 are detected using Western blotting. An asterisk denotes a non-specific band. Line graphs show the abundance of Aub- and Ago3-bound piRNA in the control (blue) and *tej* mutant (red) ovaries, via the nucleotide length. Each read number is normalized to that of small RNAs excluding piRNAs. Source data are available for this figure: SourceData FS4.

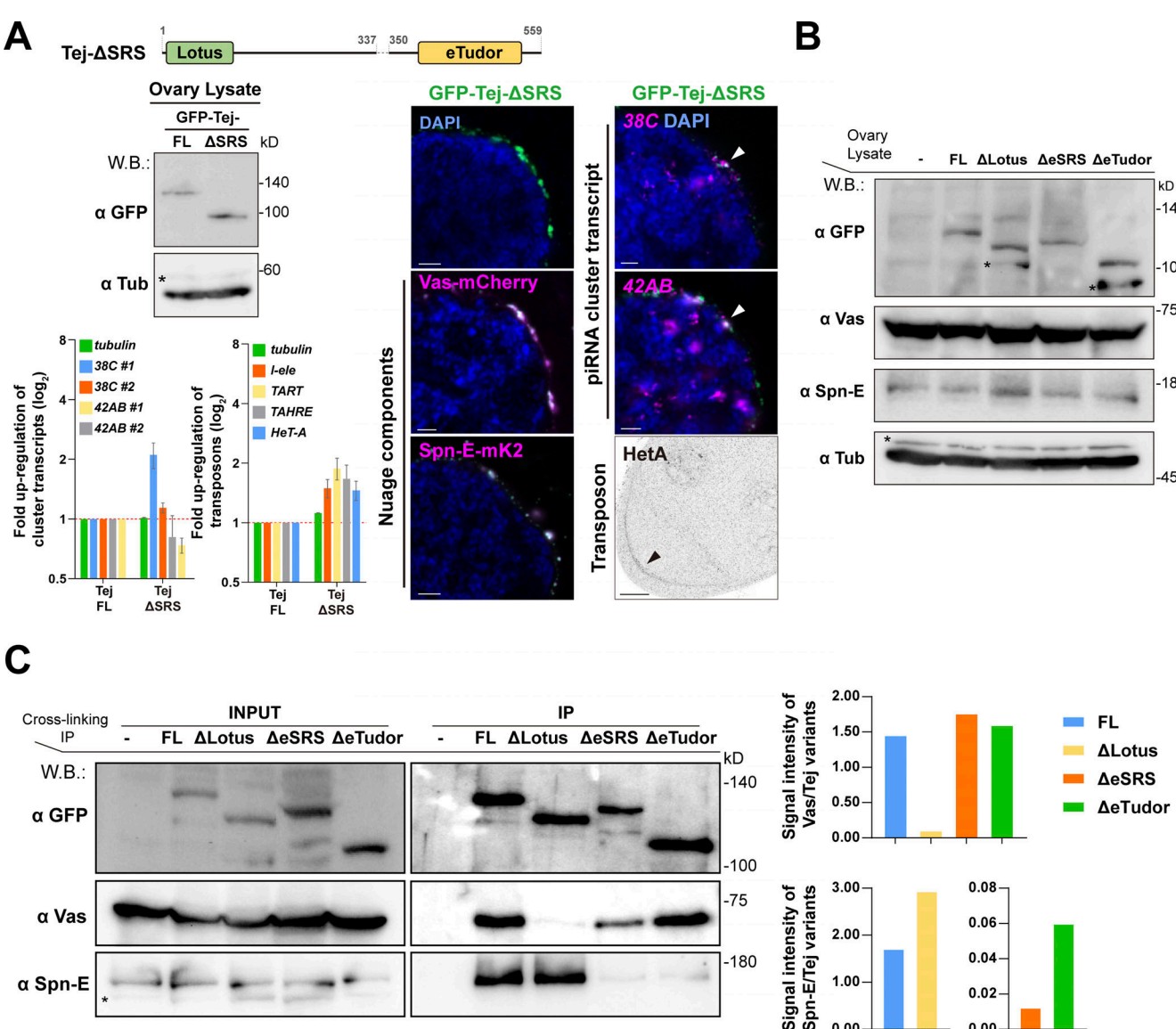

**Figure S5.   Ovaries expressing Tej variants and the genetic hierarchy among Tej, Spn-E, and Vas. (A)** Images of ovaries expressing GFP-Tej-ΔSRS with Vas-mCherry and Spn-E-mK2 (magenta, right panels). Western blot shows the expression level of Tej-FL and Tej-ΔSRS in *tej* mutant ovaries (*tej^48–5*), asterisks denote non-specific bands (left). HCR-FISH detects cluster *38C* and *42AB* transcripts (magenta, right panels), and immunostaining shows the HeT-A Gap protein (right, bottom panel, black arrowhead) in *tej* mutant (*tej^48–5*) germline cells expressing Tej-ΔSRS. The bar graph shows the fold changes of the piRNA cluster *38C* and *42AB* transcripts (top), and transposon transcripts, *I-element*, *TART*, *TAHRE*, and *HeT-A* (bottom), with *tubulin* as a control. All values are normalized to *rp49* and shown as relative expression levels compared to that in the ovaries expressing Tej-FL. Error bars indicate standard deviation (*n* = 3, number of analyzed independent experiments). **(B)** Western blot shows the expression levels of the GFP-Tej variants from transgenes, and endogenous Vas, Spn-E, and tubulin in the ovaries. Asterisks denote non-specific bands. **(C)** Immunoprecipitants of the Tej variants expressing *tej* mutant (*tej^48–5*) ovaries were detected by Western blotting with GFP, Vas, and Spn-E. Vas and Spn-E are associated with Tej via individual unique motifs. Asterisks denote non-specific bands. Densitometry analyses of the Western blotting results (left) are shown in bar graphs (right). Normalized signal intensity of Vas by Tej (top, right) and Spn-E by Tej (bottom, right) for each condition. Source data are available for this figure: SourceData FS5.

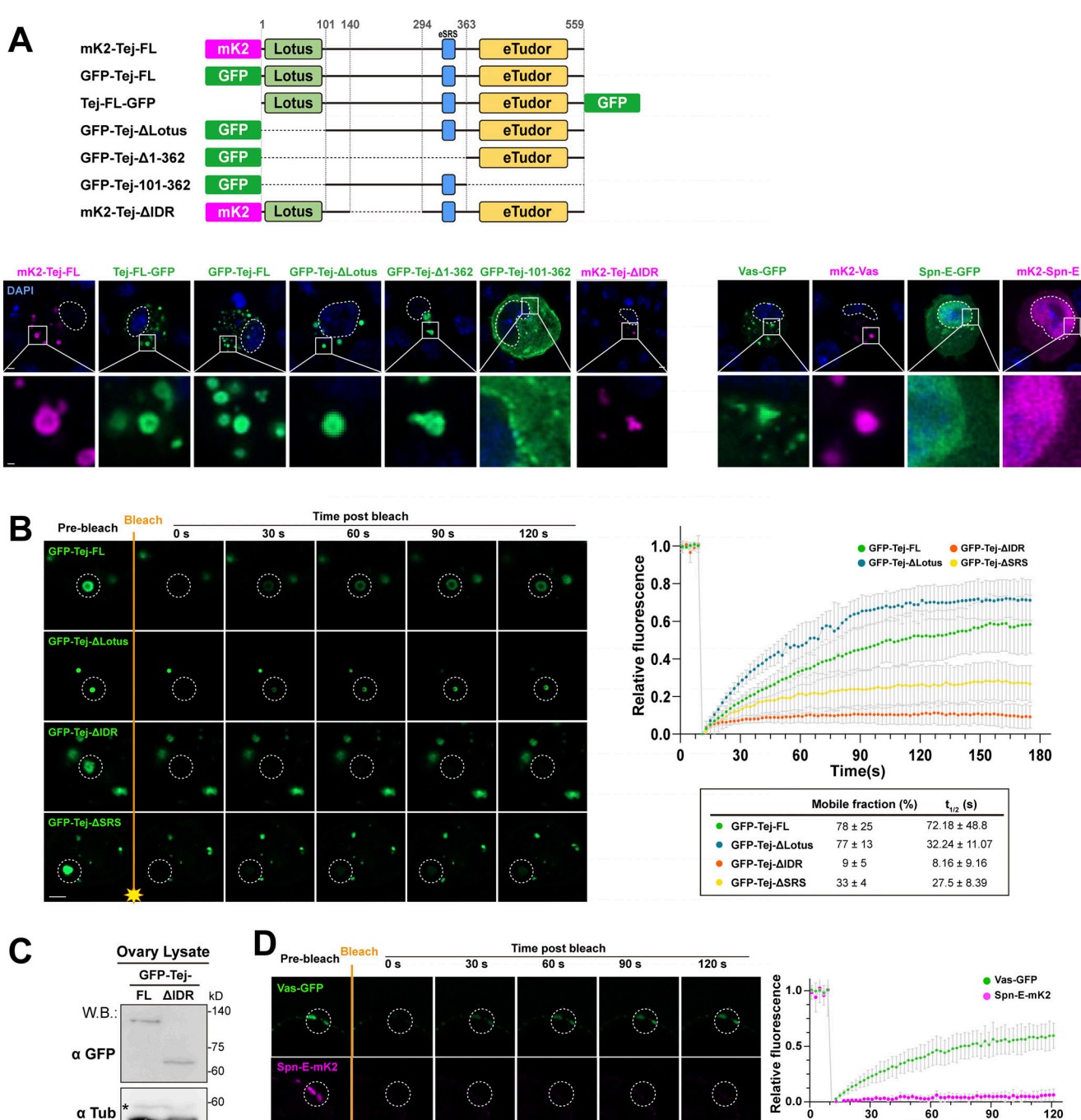

Figure S6. **Particular domains of Tej control the morphology and the mobility of Tej-formed aggregations. (A)** Schematic drawings represent the fluorophore fused proteins (top). GFP- or mK2- tagged Tej variants, Vas and Spn-E at either N- or C-terminus are expressed in S2 cells. The DNA is stained with DAPI (blue). Enlarged images of granules are shown at the bottom. **(B)** Fluorescent recovery after the photobleaching of each single granule (dotted circles) in GFP-Tej-FL or each indicated variant (green) in S2 cells. The line graph shows the normalized relative fluorescence recovery rate. The mean and ± SD are represented by colored dots and gray bars, respectively (*n* = 3, number of analyzed independent experiments). The proportion of the mobile fraction and $t_{1/2}$ derived from the mean value of the fitting curves are shown in the table. **(C)** Western blot shows the expression level of Tej-FL and Tej-ΔIDR in *tej* mutant ovaries (*tej*[48–5]). Asterisks denote non-specific bands. **(D)** The mobility of Vas-GFP or Spn-E-mK2, in the ovaries, is observed by FRAP. Vas shows higher mobility than Spn-E in vivo. The images show the recovery of the fluorescent signals for the aggregates before and after photobleaching (dotted white circles). The line graph shows the normalized relative recovery rate (bottom panel). The mean and ±SD are represented by colored dots and gray bars, respectively (*n* = 3, number of analyzed independent experiments). Scale bars, 2 µm (A, B, and D), 0.4 µm (A, lower panels). Source data are available for this figure: SourceData FS6.

**Provided online are Table S1, Table S2, and Table S3. Table S1 lists *Drosophila* genotypes used in this study. Table S2 lists antibodies used in this study. Table S3 lists primers used for qRT-PCR in this study.**

