## [Peer Review File · The Journal of Cell Biology]

Tejas functions as a core component in nuage and precursor processing in *Drosophila* piRNA biogenesis

Yuxuan Lin, Ritsuko Suyama, Shinichi Kawaguchi, Taichiro Iki, and Toshie Kai

Corresponding Author(s): Toshie Kai, Osaka University

Review Timeline:

Submission Date:	2023-03-27
Editorial Decision:	2023-04-26
Revision Received:	2023-06-11
Editorial Decision:	2023-07-05
Revision Received:	2023-07-12

Monitoring Editor: Julius Brennecke

Scientific Editor: Dan Simon

Transaction Report:

DOI: <https://doi.org/10.1083/jcb.202303125>

April 26, 2023

Re: JCB manuscript #202303125

Dr. Toshie Kai
Germline Biology Group, Integrated Biology Laboratories, Graduate School of Frontier Biosciences, Osaka University
Osaka 565-0871
Japan

Dear Dr. Kai,

Thank you for submitting your manuscript entitled "Tejas functions as a core component in nuage and precursor processing in *Drosophila* piRNA biogenesis." The manuscript was assessed by expert reviewers, whose comments are appended to this letter. We invite you to submit a revision if you can address the reviewers' key concerns, as outlined here.

You will see that the Reviewers are enthusiastic about your work and feel that it provides an important advance in our understanding of piRNA biogenesis. We agree with Reviewer #3 that testing whether Tej and Spn-E bind each other directly and quantifying microscopy-based experiments are important and should be done. Reviewers #1&3 note that the results regarding LLPS and condensate formation are not definitive and thus these claims should be toned down, although we will welcome new data expanding on this aspect if you choose to include it. The requests for changes to text and figures to better describe results, clarify interpretation, and improve presentation should be addressed in full as well.

GENERAL GUIDELINES:

Text limits: Character count for an Article is < 40,000, not including spaces. Count includes title page, abstract, introduction, results, discussion, and acknowledgments. Count does not include materials and methods, figure legends, references, tables, or supplemental legends.

Figures: Articles may have up to 10 main text figures. Figures must be prepared according to the policies outlined in our Instructions to Authors, under Data Presentation, <https://jcb.rupress.org/site/misc/ifora.xhtml>. All figures in accepted manuscripts will be screened prior to publication.

Supplemental information: There are strict limits on the allowable amount of supplemental data. Articles may have up to 5 supplemental figures. Up to 10 supplemental videos or flash animations are allowed. A summary of all supplemental material should appear at the end of the Materials and methods section.

Please note that JCB now requires authors to submit Source Data used to generate figures containing gels and Western blots with all revised manuscripts. This Source Data consists of fully uncropped and unprocessed images for each gel/blot displayed in the main and supplemental figures. Since your paper includes cropped gel and/or blot images, please be sure to provide one Source Data file for each figure that contains gels and/or blots along with your revised manuscript files. File names for Source Data figures should be alphanumeric without any spaces or special characters (i.e., SourceDataF#, where F# refers to the associated main figure number or SourceDataFS# for those associated with Supplementary figures). The lanes of the gels/blots should be labeled as they are in the associated figure, the place where cropping was applied should be marked (with a box), and molecular weight/size standards should be labeled wherever possible. Source Data files will be made available to reviewers during evaluation of revised manuscripts and, if your paper is eventually published in JCB, the files will be directly linked to specific figures in the published article.

The typical timeframe for revisions is three to four months. While most universities and institutes have reopened labs and allowed researchers to begin working at nearly pre-pandemic levels, we at JCB realize that the lingering effects of the COVID-19 pandemic may still be impacting some aspects of your work, including the acquisition of equipment and reagents. Therefore,

if you anticipate any difficulties in meeting this aforementioned revision time limit, please contact us and we can work with you to find an appropriate time frame for resubmission. Please note that papers are generally considered through only one revision cycle, so any revised manuscript will likely be either accepted or rejected.

Thank you for this interesting contribution to Journal of Cell Biology. You can contact us at the journal office with any questions, cellbio@rockefeller.edu or call (212) 327-8588.

Sincerely,

Julius Brennecke, PhD
Monitoring Editor
Journal of Cell Biology

Dan Simon, PhD
Scientific Editor
Journal of Cell Biology

Reviewer #1 (Comments to the Authors (Required)):

In the present study, the authors describe the interplay between Tejas, SpindleE and Vasa in detail. They show that Tejas interact with Vasa and SpnE using distinct domains. Tejas binds Vasa using its Lotus domain, similar like Vasa interacts with Oskar. Moreover, they identify a region upstream of the Tejas eTUDOR domain that binds SpnE. Using AF2 predictions, the author narrow down the interacting regions within the two proteins. In the last part they assess the phase separation/condensation properties of Tejas and the effect on the Vasa and SpnE localization.

The paper has a number of interesting findings that are based on solid experiments. Especially deciphering the interplay between Vasa, SpnE and Tejas is very interesting and also the discovery of the SRS motif. However, I feel that the last part that addresses the condensation phenomenon currently appears like an add-on that for me does not natively flow with the previous sections of the manuscript. It is also not clear, why they authors focus exactly on these three components for the condensate formation. For example, did the authors look at requirement of Tejas to bind PIWI protein for nuage formation? And did the authors look at Tapas, which shares several similarities with Tejas? My recommendation for the condensation part would be to better discuss the results to provide context and also alternative explanations that could account for the observations.

In several areas the manuscript would benefit by working on phrasings and formulations so that it becomes easier to read/follow. For example:

Lines 102/103; Tej devoid of the Lotus domain (Tej- Δ Lotus) formed cytoplasmic aggregates that failed to co-localize with Vas but remained colocalized with Spn-E (Fig. 2B)45.

Here the authors cite the Jeske/Ephrussi Paper, but they need to provide more background why they cite this paper and how the findings align with their experiments etc.

Specific comments:

Vasa and SpnE do not colocalize in the absence of Tejas but IP on Vasa does not recover SpnE in the CL IP? Is this due to the fact that these components are found in vastly different numbers within the cell?

Similarly, along the same lines: In Figure 4d, SpnE still colocalized with Tejas lacking the eTUDOR domain, but in the CL-IP experiment in S4C it looks like that deletion of the SRS and the eTUDOR domain both abrogate SpnE binding. Why do the authors observe

I highly appreciate the use of AlphaFold2 (AF2) to model the interaction between SpindleE and Tejas. But I do have a few comments how to improve this part.

It is important to give the reader a feeling of how confident the model is. AF2 usually provides five different models. In the best case, these models are almost identical/very similar and the five models converge. Here, the authors have only showed the best model. All five predictions and the including PAE plots should be shown - even if not all the five models are identical. The authors anyway validate the prediction/interaction with mutations from both interaction partners that support them.

The figure that displays the structure/interaction between Tejas/SpnE is too complex and shows too many residues. I would focus

on the region that clearly shows/highlights the residues that were mutated.

When the authors switch to the investigation to the condensation part of the manuscript and the IDR region of Tejas, there is a quite strong change in the story. Neither do the authors explain what a IUPRED score is, neither do they clearly explain why they investigate the condensation phenomenon. A better explanation of their motivation would be helpful.

I also do not fully understand what the authors mean with the sentence: "whether Tej is employed in the propensities equivalent liquid-liquid phase separation (LLPS)." (lines 194/195). Then the authors refer to a number of Figures and Panels without further explanation of what is shown there. For example, Figure 2B, S2A and S5A are all large panel figures and without better guidance of the reader, it is difficult to follow the authors thoughts.

I agree with the core/shell architecture that the authors observe. The use of hexane diol should be interpreted carefully, as it was also shown to inhibit kinase/phosphatase activities.
<https://pubmed.ncbi.nlm.nih.gov/33814344/>

The authors showed here, that Vasa binding required the Tejas Lotus domain. In analogy to the Vasa/Oskar interaction, one might expect that the LOTUS/helicase interaction is direct; The question is how does phase separation fit into the picture? Vasa has been shown to be phosphorylated - could it be possible that the Vasa/Tejas interaction is modulated by phosphorylation? It should at least be mentioned that localization changes could also be a consequence of changes of the phosphorylation status of the involved components.

It seems that the eTUDOR domain is more important to condensate formation than the IDR.

The contribution of the Tejas IDR itself to condensate formation is unclear to me, because the delta IDR construct still forms condensates that colocalize with Vasa. Could it be that Tejas is a condensate client that then recruits Vasa via the LOTUS domain?

Moreover, the authors mention the importance sDMA in condensation biology. The importance of the eTudor domain could argue that the interaction with PIWI proteins through the eTUDOR is a main reason for condensation formation. But this could be easily tested.

The FRAP analysis is interesting but it challenging to derive clear, mechanistic conclusion from those experiments, because the understanding of the condensate composition is currently missing. While the authors show that Tejas interacts through the Lotus domain with Vasa and via the SRS domain with SpnE, the distribution within the condensate only suggests that SpnE is present in the center and Tejas and Vasa in the periphery. If Tejas would act as an interaction hub bringing Vasa and SpnE together, I would expect a different picture/distribution. If Tejas is one of the key components that modulate the condensate properties, then I would expect that deletion of the IDR should affect Vasa and SpnE similarly, as both bind directly to Tejas via distinct regions. Could the authors comment on this in the discussion section?

I was surprised about the levels of mobile fractions that were derived from the data.

For example, in Figure S5B: The mobile fraction of the yellow curve is estimated to be 55% even though the curves plateau at around 30%. Also the green and the yellow curve have the same initial recovery kinetics but then deviate after 30 sec.

I also have a technical question: the authors fit the data to a double term exponential equation for calculation of the half time. Does $t_{1/2}$ refer to the overall half-time of both processes? This should be explained in the material and method section. Similarly, the quality of the curve fits should be shown.

Usually, it is recommended to measure $5 \times t_{1/2}$ or better $10 \times$. I acknowledge that this is not always possible but it would be great to point how long the measurement times have been in material and methods section.

To show that Vasa/SpnE mobility is affected by Tejas - the authors could delete the Tejas Lotus domain or make use of the Tej mutant that does not bind SpnE anymore.

Minor comments:

Lines 40/41: Vasa and SpindleE are different helicases. Vasa is a DEAD box and SpindleE and DEAH box helicase, this should be corrected in the text.

Throughout the text the author use Tudor domain. Tejas has extended Tudor (eTUDOR), which are different from Tudor domain only. This should be changed to avoid confusion with the normal Tudor domains.

It would have been nice to include the deltaIDR also panel S5A - this would allow comparison of all tejas constructs side-by-side.

Several Figures miss properly labeled axes.
Figure S2D: The PAE plot misses the scale.

S3A and S3B miss numbers on the y-axis

It has been noted that the best Disorder predictor seems to be the AlphaFold itself. As the authors do use AF2 for protein-protein interaction studies, they could highlight which regions of Tejas are predicted to be disordered according to AF2.

Reviewer #2 (Comments to the Authors (Required)):

This manuscript by Lin et al. performed detailed characterization of Tejas (Tej), a piRNA biogenesis factor playing roles in piRNA precursor processing and nuage formation. By revisiting localization and association of nuage components, Tej was confirmed to interact with Vas and Spn-E in separate compartments. Their dissection of these interactions identified essential domains within Tej; the new domain SRS was successfully identified as the domain required for interaction with Spn-E. Then Tej function in piRNA precursor processing was confirmed by observing defective piRNA production and upregulation of transposons in the absence of Tej. Rescue experiments confirmed the specific domains of Tej for proper localization of nuage components and for piRNA biogenesis. Lastly, specific region of Tej was discovered to regulate nuage mobility by phase separation. These Tej characterizations are based on great combinations of experiments using knock-in/transgenic fly constructs as well as S2 cells, super-resolution confocal microscopy, crosslinking-immunoprecipitation, sequencing, in-situ hybridization, structure prediction, FRAP, and so on. The manuscript is well written, and advancement of our knowledge is clearly stated. Although Tej's requirement for piRNA biogenesis has been already shown by previous studies, more detailed characterization, discovery of new domain, and additional insights into piRNA biogenesis would make this study intriguing for readers of this journal.

Reviewer #3 (Comments to the Authors (Required)):

The manuscript by Yuxuan addresses the sub-cellular organization of the piRNA pathway in *Drosophila*. It is well known that certain factors are concentrated in so-called 'germ granules' (or nuage) around the nuclear periphery, and that these granules most likely form through processes related to liquid-liquid-phase-separation (LLPS). While LLPS is typically thought of in terms of multivalent, weak interactions, 'classical' protein-protein interactions are most likely also involved in forming the germ granules. The manuscript under evaluation dissects functional regions of the protein Tejas and how it interacts with two other well-known piRNA pathway factors: Vasa and SpnE. In addition, the authors look at piRNA precursor transcripts as well as effects on mature piRNAs in various mutants.

The basic message of the work is that Tejas interacts with Vasa via its Lotus domain, and with SpnE via a newly identified motif, that does not seem to have a defined 3D fold. Tejas also has a Tudor domain, but this domain was not found to interact with Vasa or SpnE, but was found to drive Tejas granule formation. The authors also identify an intrinsically disordered region (IDR) within Tejas which increases protein mobility within the granules, particularly that of Tejas itself and Vasa.

This is interesting work, and the results are very significant for the piRNA field in general, but also to readers interested in membraneless organelles in general. As such I am very supportive of accepting this manuscript for JCB. However, a number of issues will need to be addressed.

- 1) Overall, the microscopy-based experiments are in need of quantification. Statements of co-localizing signals, accumulation of foci close to the nuclear periphery etc. all need to be substantiated through quantification. In some cases this was done, for instance in Figure 3E, but in most cases only one or maybe two images are shown. This makes it impossible to judge how variable the observation is. This is particularly important for Figure 4 (and the text in lines 177-191).
- 2) Figure S2E/F: are the mutated residues in the interface with Tej? It was not clear to me how these were chosen. Ideally, the C-term domain of SpnE would be tested against the Tej region in a recombinant protein setting, to show it is direct. Has this been tried? I do not mean to push the authors down a road of close-to-impossible protein purifications, but if not, in our experience AF-defined domains tend to express rather well, it may be worth trying to test the predicted interaction in this way.
- 3) The hexane diol result is intriguing. Typically, people see dissolution of the whole granule, but that is often in higher %-age. I am not sure how this can be interpreted, though. It is a very toxic compound in fact, and a widely shared opinion in the field is that hexane diol experiments are not very informative. It surely cannot demonstrate if a granule forms by LLPS or not. So, please write this part a bit more careful. The results is interesting, and shows, I guess, a difference in type of interactions between Vasa and Tej versus SpnE and Tej.
- 4) The FRAP experiments are nice. However, when describing SpnE's lack of mobility, this cannot be described as a stable granule. All it shows is that SpnE is rather static (which is an interesting difference compared to Vasa).
- 5) Line 278: the authors write 'mutually exclusive binding'. This refers to Vasa and SpnE binding to Tej. The authors did not show this. They demonstrate both bind to a different region, so in fact they may bind simultaneously. The fact that Vasa and SpnE levels can be separated in mutants (Fig1) does not show that Vasa and SpnE bind mutually exclusively. IN fact, simultaneous binding (even if very transient) may be crucial...
- 6) A final minor point: the English of the manuscript should best be checked by a native speaker. I found some instances where the use of English was confusing (for instance Line 238-239, 258-260, 286-287).

Editor:

We thank all the reviewers for carefully considering our manuscript (#202303125) and for their helpful comments, which have allowed us to improve our study. We are also very glad to hear all the reviewers are positive about the potential publication of our manuscript in *Journal of Cell Biology*.

We are submitting a revised manuscript and providing a point-by-point response in blue to the reviewers' comments below. The comments from the reviewers have not been edited.

We would like to thank the editor and the reviewers again for this constructive review process, and we hope that it is now suitable to be published in *Journal of Cell Biology*. Thank you for your kind consideration and we look forward to hearing from you soon.

Reviewer #1 (Comments to the Authors (Required)):

In the present study, the authors describe the interplay between Tejas, SpindleE and Vasa in detail. They show that Tejas interact with Vasa and SpnE using distinct domains. Tejas binds Vasa using its Lotus domain, similar like Vasa interacts with Oskar. Moreover, they identify a region upstream of the Tejas eTUDOR domain that binds SpnE. Using AF2 predictions, the author narrow down the interacting regions within the two proteins. In the last part they assess the phase separation/condensation properties of Tejas and the effect on the Vasa and SpnE localization.

The paper has a number of interesting findings that are based on solid experiments. Especially deciphering the interplay between Vasa, SpnE and Tejas is very interesting and also the discovery of the SRS motif. However, I feel that the last part that addresses the condensation phenomenon currently appears like an add-on that for me does not natively flow with the previous sections of the manuscript. It is also not clear, why they authors focus exactly on these three components for the condensate formation. For example, did the authors look at requirement of Tejas to bind PIWI protein for nuage formation? And did the authors look at Tapas, which shares several similarities with Tejas? My recommendation for the condensation part would be to better discuss the results to provide context and also alternative explanations that could account for the observations.

We are very grateful for the reviewer's appreciation of our study. Our focus on the condensation formation of these three components stems from the observation of Tej forming hollow-shaped aggregates in S2 cells. This could be indicative of characteristics associated with liquid-liquid phase separation (LLPS) or a hydrogel-like structure, similar to the behavior of Oskar, which forms aggregates as phase-separated granules in S2 cells (Kistler, Trcek et al. 2018). Considering the membraneless organelle characteristics of nuage, we hypothesized that Tej might be a pivotal contributor to nuage assembly. Consequently, we embarked on investigating how this distinctive feature enables Tej's involvement in nuage formation. We rephrased the statement delivering the above idea in the revised manuscript.

L200-204: *Tej induced the formation of granular-like aggregates in S2 cells (Fig. 2B, Fig. S2A, S5A), implying a role of Tej in forming the membrane-less organelle, nuage. Consistent with this notion, IUPred analysis predicted an intrinsically disordered region in the middle part of Tej (Fig. 5A), prompting us to examine whether Tej is employed in the propensities equivalent phase separated granule^{53, 54}.*

We also thank the reviewer for touching on Tapas as well. As rightly pointed out by the reviewer, Tapas contains one Lotus domain and three eTudor domains, a similar domain pattern to Tejas (Patil, Anand et al. 2014). Tapas is localized to nuage and functions for piRNA pathway in a synergetic manner with Tejas. However, as described in our previous paper, its role in piRNA biogenesis is not as robust as Tejas, especially in ovaries, because the loss of Tapas alone does not cause severe malformation of nuage and piRNA production. Hence, Tapas plays a supplementary role of nuage when Tejas is present. Therefore we consider the molecular function of Tapas in the nuage formation is out of scope in this study, although it warrants our further understanding in the future.

In addition, though we showed Tej made up a complex with Ago3 and Aub in the ovaries (Fig. 1D), we did not examine further whether their binding is direct and needed for nuage formation. However, in this manuscript, we focused on Tej in recruiting Vas and Spn-E and properly engaging piRNA precursors in nuage.

In several areas the manuscript would benefit by working on phrasings and formulations so that it becomes easier to read/follow. For example: Lines 102/103; Tej devoid of the Lotus domain (Tej- Δ Lotus) formed cytoplasmic aggregates that failed to co-localize with Vas but remained colocalized with Spn-E (Fig. 2B)⁴⁵.

Thank you for pointing this out. Indeed, this was also pointed out by the reviewer below to clarify the reason why we cited the paper by (Jeske, Müller et al. 2017). Together with that point, as below, we rephrased and amended some sentences including those pointed by the reviewer #3.

L102-103→L105-107: *Tej lacking the Lotus domain (Tej- Δ Lotus) formed cytoplasmic aggregates and exhibited co-localization with Spn-E (Fig. 2B). Consistent with the findings of a previous study⁴⁵, Tej did not co-localize with Vas.*

L238-239→L248-250: *Although its precise function has not been clarified, our findings demonstrate that Tej plays a crucial role in recruiting RNA helicases Vas and Spn-E to nuage through distinct domains, namely, Lotus and SRS.*

L258-260→L270-272: *Despite the unusual nuage granules of Tej- Δ eTudor, it mildly suppressed transposon expression (Fig. 4B, 4F). Notably, Tej- Δ eTudor displays interaction with Vas and Spn-E, albeit to lesser extent especially with Spn-E (Fig. S4C).*

L286-287→L299-300: *Thus, Tej- Δ IDR may remain co-localized with Vas and Spn-E, facilitating the processing of piRNAs (Fig. 5C).*

Here the authors cite the Jeske/Ephrussi Paper, but they need to provide more background why they cite this paper and how the findings align with their experiments etc.

Thank you for the comment. Jeske et al. reported in 2017 that the Tej interact with Vas via the Lotus domain in the Tej N-terminal moiety. Consistently, we observed that the Tej- Δ Lotus variant lost binding to Vas and the ability to form aggregation with Vas in S2 cells and in vivo systems.

To clarify this point, we amended one line as mentioned above and another sentence in Introduction as below.

L59-60: *The Lotus domain was previously reported to interact with Vas, which is required for the piRNA pathway*^{44, 45}.

Specific comments:

Vasa and SpnE do not colocalize in the absence of Tejas but IP on Vasa does not recover SpnE in the CL IP? Is this due to the fact that these components are found in vastly different numbers within the cell?

We thank the reviewer for the comments on the Vasa and Spn-E interaction with Tej. The immunostaining in Fig. 1C shows that Vas and Spn-E were no longer co-localized but segregated from each other in the absence of Tej, suggesting that Vas and Spn-E may not directly interact, but make a complex through Tej. This is also supported by the CL-IP results showing that Vas and Spn-E IP pull down a scanty amount of Spn-E and Vas, respectively (Fig. 1D). Please kindly note that we provided the same blot of Vas-GFP IP for Spn-E and Tej with the weaker contrast in this revised manuscript, because the image was too bright and the faint signal of Spn-E has disappeared in the previous manuscript. In summary, we propose these results suggest that Tej interacts with Vas and Spn-E in a different sub-compartment, though we cannot exclude a possibility that this may be due to the detection limitation of western blotting for their simultaneous and/or transient binding.

Similarly, along the same lines: In Figure 4d, SpnE still colocalized with Tejas lacking the eTUDOR domain, but in the CL-IP experiment in S4C it looks like that deletion of the SRS and the eTUDOR domain both abrogate SpnE binding. Why do the authors observe

We thank the reviewer for pointing this out. We identified the Spn-E binding site on Tej and confirmed the interaction both *in vitro* and *in vivo*. Although Spn-E still interacts Tej- Δ eTudor in CL-IP, Tej's binding to Spn-E is relatively low but higher than with Tej- Δ eSRS (Fig. S4C). We thought that the proximal region of eSRS probably influences the interaction with Tej- Δ eTudor for binding with other proteins necessary for Spn-E binding at nuage. Alternatively, loss of nuage aggregation by Tej- Δ eTudor largely reduced the Tej-Spn-E interact efficiency. Contrarily, in both S2 cells and ovaries, Tej- Δ eTudor is dispersed mainly in the cytoplasm due to the molecular nature, which makes their interaction weaker in CL-IP.

I highly appreciate the use of AlphaFold2 (AF2) to model the interaction between Spindle-E and Tejas. But I do have a few comments how to improve this part. It is important to give the reader a feeling of how confident the model is. AF2 usually provides five different

models. In the best case, these models are almost identical/very similar and the five models converge. Here, the authors have only showed the best model. All five predictions and the including PAE plots should be shown - even if not all the five models are identical. The authors anyway validate the prediction/interaction with mutations from both interaction partners that support them. The figure that displays the structure/interaction between Tejas/SpnE is too complex and shows too many residues. I would focus on the region that clearly shows/highlights the residues that were mutated.

Thank you for your comments. Indeed, all the Rank 0-4 and the corresponding PAE plots showed confidence: all 5 predicted structures had a high consistency, especially for the Tej-Spn-E interacting regions. We provided a simplified interface between Tej-Spn-E of the best-scored structures with the highlighted mutated residues on Spn-E

Rank 0="model_3_multimer_v2_pred_0": 0.6429424908619761
 Rank 1="model_1_multimer_v2_pred_0": 0.6361931429853533
 Rank 2="model_4_multimer_v2_pred_0": 0.5944704494978156
 Rank 3="model_5_multimer_v2_pred_0": 0.5943308167499957
 Rank 4="model_2_multimer_v2_pred_0": 0.5717023253643091

in Fig. S2D in the revised manuscript. PAE plots are quite similar among the five predicted structures, and we attached those superimposed structures for the reviewer for your reference as below.

The Figure shows superimposed all five predicted structures. The five predicted models are ranked with Predicted Template Modeling (pTM) score and colored by Ranking. Their corresponding PAE plots are shown in the bottom. The Rank 0 model (in orange) is partially displayed in Fig. S2D.

When the authors switch to the investigation to the condensation part of the manuscript and the IDR region of Tejas, there is a quite strong change in the story. Neither do the authors explain what a IUPRED score is, neither do they clearly explain why they investigate the condensation phenomenon. A better explanation of their motivation would be helpful.

Thank you for the comment. As to IUPred, we described in Material and Methods in the original manuscript as below.

L540-542: The intrinsically disordered region was analyzed with the IUPred server (<https://iupred2a.elte.hu/>). The region containing residues with IUPred scores more than 0.5 was classified as a prominent intrinsically disordered region^{53, 54}.

In addition, as suggested, we added the statement explaining the reason why we focus on the IDR region in the revised manuscript (same as above).

L200-204: Tej induced the formation of granular-like aggregates in S2 cells (Fig. 2B, Fig. S2A, S5A), implying a role of Tej in forming the membrane-less organelle, nuage. Consistent with this notion, IUPred analysis predicted an intrinsically disordered region in the middle part of Tej (Fig. 5A), prompting us to examine whether Tej is employed in the propensities equivalent phase separated granule^{53, 54}.

I also do not fully understand what the authors mean with the sentence: "whether Tej is employed in the propensities equivalent liquid-liquid phase separation (LLPS)." (lines 194/195). Then the authors refer to a number of Figures and Panels without further explanation of what is shown there. For example, Figure 2B, S2A and S5A are all large panel figures and without better guidance of the reader, it is difficult to follow the authors thoughts.

Thank you for the comment. We referred to these figures to show that Tej expressed in S2 cells exhibited a strong tendency to form round hollow structures, regardless of the position of fluorophores at the N- or C- terminal of Tej. To make this explanation clear, we rephrased the text in the revised manuscript as already described above (L200-204 in the revised manuscript).

I agree with the core/shell architecture that the authors observe. The use of hexane diol should be interpreted carefully, as it was also shown to inhibit kinase/phosphatase activities. <https://pubmed.ncbi.nlm.nih.gov/33814344/>
The authors showed here, that Vasa binding required the Tejas Lotus domain. In analogy to the Vasa/Oskar interaction, one might expect that the LOTUS/helicase interaction is

direct; The question is how does phase separation fit into the picture? Vasa has been shown to be phosphorylated - could it be possible that the Vasa/Tejas interaction is modulated by phosphorylation? It should at least be mentioned that localization changes could also be a consequence of changes of the phosphorylation status of the involved components.

Thank you for your comment. Although we have no experimental evidence to exclude this possibility, phosphorylation of Vas is a potential factor affecting the interaction with other proteins. We added this notion and a suggested reference below in our revised manuscript.

L305-307: We also cannot exclude the possibility that 1,6-HD treatment might have impaired kinase and/or phosphatase activity⁶³. Hence localization might have been affected by the changes in their phosphorylation status.

It seems that the eTUDOR domain is more important to condensate formation than the IDR. The contribution of the Tejas IDR itself to condensate formation is unclear to me, because the delta IDR construct still forms condensates that colocalize with Vasa. Could it be that Tejas is a condensate client that then recruits Vasa via the LOTUS domain? Moreover, the authors mention the importance sDMA in condensation biology. The importance of the eTudor domain could argue that the interaction with PIWI proteins through the eTUDOR is a main reason for condensation formation. But this could be easily tested.

Thank you for the comments about the condensates structure of Tej. As it induced the aggregation of Tej, eTudor is more critical than IDR, yes, we also think Spn-E and Tej can recruit Vas. As rightly pointed out, eTudor domain is known to recognize and interact with sDMA on PIWI proteins. However, our functional analysis in this study propose that Tej plays a dominant role in piRNA precursor recruitment to nuage together with Vas and Spn-E, prior to the processing of precursors by PIWI family proteins at nuage, as we mentioned above. Hence, in this study, we focus on the roles of Tej, Vas and Spn-E and their interaction, but not PIWI family proteins, in the aspect of condensation biology.

The FRAP analysis is interesting but it challenging to derive clear, mechanistic conclusion from those experiments, because the understanding of the condensate composition is currently missing. While the authors show that Tejas interacts through the Lotus domain with Vasa and via the SRS domain with SpnE, the distribution within the condensate only suggests that SpnE is present in the center and Tejas and Vasa in the periphery. If Tejas would act as an interaction hub bringing Vasa and SpnE together, I would expect a different picture/distribution. If Tejas is one of the key components that modulate the condensate properties, then I would expect that deletion of the IDR should affect Vasa and SpnE similarly, as both bind directly to Tejas via distinct regions. Could the authors comment on this in the discussion section?

We appreciate the reviewer's comments on this issue. In S2 cells, the association of Vas or Spn-E to Tej was not changed by the loss of IDR of Tej (Fig. 5C). This is consistent with our findings that Tej interacts with Vas and Spn-E via the Lotus domain and SRS, respectively (Fig.S4C). However, Vas mobility was attenuated by loss of IDR in the FRAP analysis (Fig. 5D), and the treatment of 1,6-HD affected Vas

distribution (Fig. 5B). In contrast, Spn-E mobility was not affected by the loss of IDR of Tej, and the distribution was not changed by the treatment of 1,6-HD (Fig. 5B, 5D), implying that Spn-E interacts Tej relatively rigid among these proteins. Taken together, these results suggest that the interaction between Tej with Vas, but not with Spn-E, can be affected through the hydrophobic interaction or IDR of Tej.

We provided this argument in Discussion of the revised manuscript as below.

L307-310: The behavior of these proteins is seemingly influenced by their respective binding modes and properties with Tej. The interaction of Vas with Tej is affected by 1,6-HD and IDR region of Tej through the hydrophobic association, whereas that of Spn-E with Tej is more rigid, possibly contributing to form the scaffold of nuage.

I was surprised about the levels of mobile fractions that were derived from the data. For example, in Figure S5B: The mobile fraction of the yellow curve is estimated to be 55% even though the curves plateau at around 30%. Also the green and the yellow curve have the same initial recovery kinetics but then deviate after 30 sec.

I also have a technical question: the authors fit the data to a double term exponential equation for calculation of the half time. Does $t_{1/2}$ refer to the overall half-time of both processes? This should be explained in the material and method section. Similarly, the quality of the curve fits should be shown. Usually, it is recommended to measure $5 \times t_{1/2}$ or better $10 \times$. I acknowledge that this is not always possible but it would be great to point how long the measurement times have been in material and methods section.

We thank the reviewer for asking about analysis of FRAP. We used easyFRAP (Rapsomaniki, Kotsantis et al. 2012), an online full-scale normalization procedure for FRAP measurement to correct for differences in bleaching depth among each experiment. All normalized recovery curves were then fitted with a double-term exponential equation on web-based methods. The half-time and mobile fraction refers to the mean fitting curve. The actual measurement times for the data in Fig. S5B was 170s, longer observation might represent a higher curve plateau.

Following your suggestion, we added the above description in the Methods and Materials section.

L536-538: Individual normalized data were fitted with a double-term exponential equation and used for calculation of the half-time of full fluorescence recovery ($t_{1/2}[s]$) and the percentage of mobile fraction. Each value is averaged and represented in the table.

We also appreciate the reviewer for asking about the levels of mobile fractions. We re-checked the analysis data upon the suggestion and realized a few individual fitting curve out of multiple repetitions of measurement are not convergent. It caused misleading the data in the previous version in the averages of the levels of mobile fractions. We recalculated carefully and corrected the levels of mobile fraction deviated by recovering the fluorescent intensity in the measurement time in Fig. 5D, Fig. 5G, and Fig. S5B. Also we changed the values, accordingly in the text.

L228-233: FRAP experiments showed that more than 80% of Vas was recovered within 90 s with Tej-FL ($t_{1/2}$: 23.16 ± 11.46 s); however less than 25% was slowly recovered with Tej- Δ IDR ($t_{1/2}$: 59.28 ± 40.73 s), indicating that Tej IDR facilitated the mobility of Vas (Fig. 5D, upper panel). However, less than 40% of Spn-E was slowly recovered with Tej-FL or Tej- Δ IDR (Fig. 5D, lower panel), indicating Spn-E formed rather static granules.

L238-243: Notably, the mobility of Vas in nuage was more attenuated by Tej- Δ IDR than by Tej-FL (recovery rate: 82%, $t_{1/2}$: 10.79 s with Tej-FL vs 62%, $t_{1/2}$: 77.35 s with Tej- Δ IDR; Fig. 5G). In conclusion, these results suggest that Tej IDR contributes to the mobility of nuage components not only that of Tej itself but also that of Vas, while Spn-E remains relatively static.

To show that Vasa/SpnE mobility is affected by Tejas - the authors could delete the Tejas Lotus domain or make use of the Tej mutant that does not bind SpnE anymore.

Thank you for the comment. We wished to know the mobility of interacting proteins by the influence of Tej. However, Tej devoid of Lotus domain does not interact with Vas nor colocalized with Vas (Fig. 2B, 4B, Fig. S4C). Therefore, we consider Vas mobility under this condition out of our scope.

As to Spn-E, upon loss of SRS/eSRS of Tej, Spn-E is delocalized from nuage in vivo (Fig. 4B), does not co-localize with Tej rather dispersed in S2 cell nuclei (Fig. 2B). Moreover, we revealed Spn-E does not interact with Tej- Δ SRS (Fig. S4C). Hence it is impossible to examine Spn-E mobility by FRAP analysis.

Minor comments:

Lines 40/41: Vasa and SpindleE are different helicases. Vasa is a DEAD box and SpindleE and DEAH box helicase, this should be corrected in the text.

Thanks for pointing this out. We have rephrased this sentence according to your suggestion in the revised manuscript.

L40-44: *Nuage consists of precursors and transposon RNAs being processed, two PIWI family proteins—Aub and Ago3—and other relevant components, DEAD-box RNA helicase Vasa (Vas), DEAH box helicase RNA helicase Spindle-E (Spn-E), and a group of Tudor domain-containing proteins (Tdrds), Krimper (Krimp), Tejas (Tej), Tudor, Tapas, Qin/Kumo, and Vreteno^{5, 22-30}.*

Throughout the text the author use Tudor domain. Tejas has extended Tudor (eTUDOR), which are different from Tudor domain only. This should be changed to avoid confusion with the normal Tudor domains.

We have introduced extended Tudor and changed all “Tudor” to “eTudor” throughout this revised manuscript.

It would have been nice to include the deltaIDR also panel S5A - this would allow comparison of all tejas constructs side-by-side.

We included the schematic drawing for delta IDR to the Fig. S5A of the revised manuscript.

Several Figures miss properly labeled axes. Figure S2D: The PAE plot misses the scale. S3A and S3B miss numbers on the y-axis

We have added the scales in Fig. S2D and numbers on the y-axes in Fig. S3A and S3B in the revised manuscript.

It has been noted that the best Disorder predictor seems to be the AlphaFold itself. As the authors do use AF2 for protein-protein interaction studies, they could highlight which regions of Tejas are predicted to be disordered according to AF2.

We replaced the prediction result of the Tej-Spn-E structure in Fig. S2D with the one predicted using the version of AF2.2, since the previous figure was used in AF2.1. We show the IDR of Tej clearly, though it is truncated due to space limitations. Also, we attached the five different predicted structures in this letter, as mentioned above.

We added these results in Fig. S2D in the revised manuscript and accordingly amended the legend as below.

L999-1004: The predicted interface between Tej and Spn-E using AlphaFold v2.2 indicates the 91-493 aa of Spn-E and the full length of Tej (left). The inset shows an enlarged view of the interface of Tej and Spn-E interaction (right bottom). The numbered residues in Spn-E (blue), representing those predicted to interact with eSRS (red), were mutated in the subsequent experiments. The predicted aligned error plot between Tej and Spn-E is displayed (right top).

Reviewer #2 (Comments to the Authors (Required)):

This manuscript by Lin et al. performed detailed characterization of Tejas (Tej), a piRNA biogenesis factor playing roles in piRNA precursor processing and nuage formation. By revisiting localization and association of nuage components, Tej was confirmed to interact with Vas and Spn-E in separate compartments. Their dissection of these interactions identified essential domains within Tej; the new domain SRS was successfully identified as the domain required for interaction with Spn-E. Then Tej function in piRNA precursor processing was confirmed by observing defective piRNA production and upregulation of transposons in the absence of Tej. Rescue experiments confirmed the specific domains of Tej for proper localization of nuage components and for piRNA biogenesis. Lastly, specific region of Tej was discovered to regulate nuage mobility by phase separation. These Tej characterizations are based on great combinations of experiments using knock-in/transgenic fly constructs as well as S2 cells, super-resolution confocal microscopy, crosslinking-immunoprecipitation, sequencing, in-situ hybridization, structure prediction, FRAP, and so on. The manuscript is well written, and advancement of our knowledge is clearly stated. Although Tej's requirement for piRNA biogenesis has been already shown by previous studies, more detailed characterization, discovery of new domain, and additional insights into piRNA biogenesis would make this study intriguing for readers of this journal.

We are grateful for the comments and appreciation for our study by the reviewer.

Reviewer #3 (Comments to the Authors (Required)):

The manuscript by Yuxuan addresses the sub-cellular organization of the piRNA pathway in *Drosophila*. It is well known that certain factors are concentrated in so-called 'germ granules' (or nuage) around the nuclear periphery, and that these granules most likely form through processes related to liquid-liquid-phase-separation (LLPS). While LLPS is typically thought of in terms of multivalent, weak interactions, 'classical' protein-protein interactions are most likely also involved in forming the germ granules. The manuscript under evaluation dissects functional regions of the protein Tejas and how it interacts with two other well-known piRNA pathway factors: Vasa and SpnE. In addition, the authors look at piRNA precursor transcripts as well as effects on mature piRNAs in various mutants.

The basic message of the work is that Tejas interacts with Vasa via its Lotus domain, and with SpnE via a newly identified motif, that does not seem to have a defined 3D fold. Tejas also has a Tudor domain, but this domain was not found to interact with Vasa or SpnE, but was found to drive Tejas granule formation. The authors also identify an intrinsically disordered region (IDR) within Tejas which increases protein mobility within the granules, particularly that of Tejas itself and Vasa.

This is interesting work, and the results are very significant for the piRNA field in general, but also to readers interested in membraneless organelles in general. As such I am very supportive of accepting this manuscript for JCB. However, a number of issues will need to be addressed.

1) Overall, the microscopy-based experiments are in need of quantification. Statements of co-localizing signals, accumulation of foci close to the nuclear periphery etc. all need to be substantiated through quantification. In some cases this was done, for instance in Figure 3E, but in most cases only one or maybe two images are shown. This makes it impossible to judge how variable the observation is. This is particularly important for Figure 4 (and the text in lines 177-191).

Thank you for pointing this out. As suggested, we show the results of quantified piRNA cluster transcripts as in situ foci in the proximity region to the nuclear periphery with

the same methods in Fig. 3E. We added these results in Fig. 4C in the revised manuscript and accordingly amended the text with the quantified numbers and the legend.

L182-184: *We further examined the localization of piRNA cluster transcripts upon the expression of each Tej variant in tej mutant germline cells and quantified the foci of the cluster transcripts of 38C or 42AB in the perinuclear region (Fig. 4C).*

L924-928: *The ratio of the fluorescence intensities of the piRNA precursors in the nuclear membrane vicinity of Tej variants rescued germline cells. The signal intensity of the foci located inside and outside the nuclear membrane within a distance of 5% of the nucleus diameter is quantified (n=10), normalized with that in the nucleus, and plotted as percentiles relative to the total intensity.*

2) Figure S2E/F: are the mutated residues in the interface with Tej? It was not clear to me how these were chosen. Ideally, the C-term domain of SpnE would be tested against the Tej region in a recombinant protein setting, to show it is direct. Has this been tried? I do not mean to push the authors down a road of close-to-impossible protein purifications, but if not, in our experience AF-defined domains tend to express rather well, it may be worth trying to test the predicted interaction in this way.

Thank you for your comments. We presented the mutated residues on Spn-E in the new Fig. S2D, and showed they are located in the two alpha-helices on the interface with Tej. In order to verify the essential part of Spn-E for interacting with Tej in vitro, we truncated Spn-E into two parts, the highly structured N-terminal (aa1-800) and the eTudor domain-containing C-terminal (aa801-1434). We tried to purify them using a His tag expression system, especially in aiming to see the direct interaction between Tej and the N-ter of Spn-E containing the CTD domain, predicted interaction region with the eSRS of Tej.

Unfortunately, the purification of bacterially expressed full-length and two truncated variants of Spn-E was unsuccessful. The full length of Spn-E failed to express, and both the N- and C-terminal of Spn-E truncated variants remained in insoluble fraction.

Thus, we ended up with abandoning to examine the direct interaction in vitro with purified proteins.

The actual gel images for the SDS-PAGE are provided above. The gels were stained with Quick-CBB plus to show the band of proteins. The red arrows mark out the actual or expected position of target peptides. The **Total** sample includes the full lysis of the E.coli cells, the **Sup** sample stands for the supernatant after sonication and **Pellet** was the insoluble precipitate.

3) The hexanediol result is intriguing. Typically, people see dissolution of the whole granule, but that is often in higher %-age. I am not sure how this can be interpreted, though. It is a very toxic compound in fact, and a widely shared opinion in the field is that hexane diol experiments are not very informative. It surely cannot demonstrate if a granule forms by LLPS or not. So, please write this part a bit more careful. The results is interesting, and shows, I guess, a difference in type of interactions between Vasa and Tej versus SpnE and Tej.

Thank you for reminding us of the shortcoming of hexanediol treatment. We performed this experiment carefully to observe the subcellular localization of testing components in S2 cells. We conducted more than three biological replicates with the appropriate control (the same volume of the solvent was added). However, we understand the concerns in the field, and did not conclude that LLPS regulated their distinct behavior in the original manuscript. Please kindly refer to the sentence below.

L206-211: We treated the cells with 1,6-hexanediol (1,6-HD), to disturb the weak hydrophobic interactions, and found that Vas significantly re-localized from the periphery to the center of the granule together with Spn-E while Tej remained at the periphery (Fig. 5B). This finding suggest that the higher mobility of Vas was intervened by the weak hydrophobic interactions among these proteins.

In addition, we rephrased the text in Discussion of the revised manuscript as below.

L292-294: Our further results with S2 revealed that the weak hydrophobic interaction between the proteins may contribute to the formation and regulation of membrane-less structures on nuage.

L300-304: Alternatively, the reduction of Vas mobility by the loss of Tej IDR could be compensated by other components in nuage. Only the localization of Vas was remarkably changed upon 1,6-HD treatment in S2 cells, further supporting the finding that weak hydrophobic interaction controlled the dynamics of Vas, although we cannot exclude a possibility of the unexpected effects by the 1,6-HD treatment.

4) The FRAP experiments are nice. However, when describing SpnE's lack of mobility, this cannot be described as a stable granule. All it shows is that SpnE is rather static (which is an interesting difference compared to Vasa).

We appreciate the comments by the reviewer. We have changed the description as suggested.

L231-233: *However, less than 40% of Spn-E was slowly recovered with Tej-FL or Tej-ΔIDR (Fig. 5D, lower panel), indicating Spn-E formed rather static granules.*

L240-243: *In conclusion, these results suggest that Tej IDR contributes to the mobility of nuage components not only that of Tej itself but also that of Vas, while Spn-E remains relatively static.*

5) Line 278: the authors write 'mutually exclusive binding'. This refers to Vasa and SpnE binding to Tej. The authors did not show this. They demonstrate both bind to a different region, so in fact they may bind simultaneously. The fact that Vasa and SpnE levels can be separated in mutants (Fig1) does not show that Vasa and SpnE bind mutually exclusively. IN fact, simultaneous binding (even if very transient) may be crucial...

Thank you for pointing this out. As rightly commented, we raised a possibility that Vasa and Spn-E binding to Tej 'mutually exclusively' in vitro. However, it cannot exclude the other possibility of their simultaneous and/or transient binding, which may be under the detection level. In addition, in this revised manuscript, we provided the same blot of Vas-GFP IP for Spn-E and Tej with the weaker contrast, because the image was too bright and the faint signal of Spn-E has disappeared in the previous manuscript. Accordingly, we deleted the text touching 'mutually exclusive interaction' and rephrased the following text in the revised manuscript below.

Original text: These two different complexes can be supported by the mutually exclusive interaction of Vas and Spn-E to Tej, suggesting Tej may allow them to compartmentalize with each other, as seen in CL-IP (Fig. 1D) and also reported in Bombyx germ cell.

Rephrased to:

L290-292: *These results suggest that Tej may facilitate the compartmentalization of Vas and Spn-E, as shown in CL-IP experiments (Fig. 1D) and also reported in Bombyx germ cells⁶⁰, while we cannot exclude the possibility of simultaneously binding among these proteins.*

6) A final minor point: the English of the manuscript should best be checked by a native speaker. I found some instances where the use of English was confusing (for instance Line 238-239, 258-260, 286-287).

Thank you for your suggestion. We have double checked our manuscript and correct the confusing descriptions including specific points.

L238-239→L248-250: *Although its precise function has not been clarified, our findings demonstrate that Tej plays a crucial role in recruiting RNA helicases Vas and Spn-E to nuage through distinct domains, namely, Lotus and SRS.*

L258-260→L270-272: *Despite the unusual nuage granules of Tej-ΔeTudor, it mildly suppressed transposon expression (Fig. 4B, 4F). Notably, Tej-ΔeTudor displays interaction with Vas and Spn-E, albeit to lesser extent especially with Spn-E (Fig. S4C).*

L286-287→L299-300: *Thus, Tej-ΔIDR may remain co-localized with Vas and Spn-E, facilitating the processing of piRNAs (Fig. 5C).*

Reference

Jeske, M., C. W. Müller and A. Ephrussi (2017). "The LOTUS domain is a conserved DEAD-box RNA helicase regulator essential for the recruitment of Vasa to the germ plasm and nuage." *Genes Dev* 31(9): 939-952.

Kistler, K. E., T. Trcek, T. R. Hurd, R. Chen, F. X. Liang, J. Sall, M. Kato and R. Lehmann (2018). "Phase transitioned nuclear Oskar promotes cell division of *Drosophila* primordial germ cells." *Elife* 7.

Patil, V. S., A. Anand, A. Chakrabarti and T. Kai (2014). "The Tudor domain protein Tapas, a homolog of the vertebrate Tdrd7, functions in the piRNA pathway to regulate retrotransposons in germline of *Drosophila melanogaster*." *BMC Biol* 12: 61.

Rapsomaniki, M. A., P. Kotsantis, I.-E. Symeonidou, N.-N. Giakoumakis, S. Taraviras and Z. Lygerou (2012). "easyFRAP: an interactive, easy-to-use tool for qualitative and quantitative analysis of FRAP data." *Bioinformatics* 28(13): 1800-1801.

July 5, 2023

RE: JCB Manuscript #202303125R

Dr. Toshie Kai
Osaka University
Germline Biology Group, Integrated Biology Laboratories, Graduate School of Frontier Biosciences, Osaka University
Osaka 565-0871
Japan

Dear Dr. Kai,

Thank you for submitting your revised manuscript entitled "Tejas functions as a core component in nuage and precursor processing in *Drosophila* piRNA biogenesis." We would be happy to publish your paper in JCB pending final text and figure revisions necessary to meet our formatting guideline as well as to address the final reviewer comments (see details below).

We agree with Reviewer #1 that it would be important to address the remaining open points, provide clarification and additional explanation where needed, and to add the AlphaFold models and PAE plots to the supplement. We do not feel that it is essential to add assays of Vasa mobility in cells where Tejas does not bind Spn-E.

A. MANUSCRIPT ORGANIZATION AND FORMATTING:

1) Text limits: Character count for Articles is < 40,000, not including spaces. Count includes title page, abstract, introduction, results, discussion, and acknowledgments. Count does not include materials and methods, figure legends, references, tables, or supplemental legends.

2) Figure formatting: Articles may have up to 10 main text figures. Scale bars must be present on all microscopy images, including inset magnifications. Molecular weight or nucleic acid size markers must be included on all gel electrophoresis. Please add scale bars to magnifications in Figures 1A/C & 5B/C as well as MW markers to blots in 1D & the top rows of S3A/B.

Also, please avoid pairing red and green for images and graphs to ensure legibility for color-blind readers. If red and green are paired for images, please ensure that the particular red and green hues used in micrographs are distinctive with any of the colorblind types. If not, please modify colors accordingly or provide separate images of the individual channels.

3) Statistical analysis: Error bars on graphic representations of numerical data must be clearly described in the figure legend. The number of independent data points (n) represented in a graph must be indicated in the legend. Please, indicate whether 'n' refers to technical or biological replicates (i.e. number of analyzed cells, samples or animals, number of independent experiments). If independent experiments with multiple biological replicates have been performed, we recommend using distribution-reproducibility SuperPlots (please see Lord et al., JCB 2020) to better display the distribution of the entire dataset, and report statistics (such as means, error bars, and P values) that address the reproducibility of the findings.

Statistical methods should be explained in full in the materials and methods. For figures presenting pooled data the statistical measure should be defined in the figure legends. Please also be sure to indicate the statistical tests used in each of your experiments (both in the figure legend itself and in a separate methods section) as well as the parameters of the test (for example, if you ran a t-test, please indicate if it was one- or two-sided, etc.). Also, if you used parametric tests, please indicate if the data distribution was tested for normality (and if so, how). If not, you must state something to the effect that "Data distribution was assumed to be normal but this was not formally tested."

4) Materials and methods: Should be comprehensive and not simply reference a previous publication for details on how an experiment was performed. Please provide full descriptions (at least in brief) in the text for readers who may not have access to referenced manuscripts. The text should not refer to methods "...as previously described." Please also indicate the acquisition and quantification methods for immunoblotting/western blots.

5) For all cell lines, vectors, constructs/cDNAs, etc. - all genetic material: please include database / vendor ID (e.g., Addgene, ATCC, etc.) or if unavailable, please briefly describe their basic genetic features, even if described in other published work or gifted to you by other investigators (and provide references where appropriate). Please be sure to provide the sequences for all

of your oligos: primers, si/shRNA, RNAi, gRNAs, etc. in the materials and methods. You must also indicate in the methods the source, species, and catalog numbers/vendor identifiers (where appropriate) for all of your antibodies, including secondary. If antibodies are not commercial, please add a reference citation if possible.

6) Microscope image acquisition: The following information must be provided about the acquisition and processing of images:

- a. Make and model of microscope
- b. Type, magnification, and numerical aperture of the objective lenses
- c. Temperature
- d. Imaging medium
- e. Fluorochromes
- f. Camera make and model
- g. Acquisition software
- h. Any software used for image processing subsequent to data acquisition. Please include details and types of operations involved (e.g., type of deconvolution, 3D reconstitutions, surface or volume rendering, gamma adjustments, etc.).

7) References: There is no limit to the number of references cited in a manuscript. References should be cited parenthetically in the text by author and year of publication. Abbreviate the names of journals according to PubMed.

8) Supplemental materials: Articles generally have up to 5 supplemental figures and 10 videos. In this case, we will be able to give you the extra space to accommodate the AlphaFold models and PAE plots. Please also note that tables, like figures, should be provided as individual, editable files. A summary of all supplemental material should appear at the end of the Materials and methods section. Please include one brief sentence per item.

9) eTOC summary: A ~40-50 word summary that describes the context and significance of the findings for a general readership should be included on the title page. The statement should be written in the present tense and refer to the work in the third person. It should begin with "First author name(s) et al..." to match our preferred style.

10) Conflict of interest statement: JCB requires inclusion of a statement in the acknowledgements regarding competing financial interests. If no competing financial interests exist, please include the following statement: "The authors declare no competing financial interests." If competing interests are declared, please follow your statement of these competing interests with the following statement: "The authors declare no further competing financial interests."

11) A separate author contribution section is required following the Acknowledgments in all research manuscripts. All authors should be mentioned and designated by their first and middle initials and full surnames. We encourage use of the CRediT nomenclature (<https://casrai.org/credit/>).

12) ORCID IDs: ORCID IDs are unique identifiers allowing researchers to create a record of their various scholarly contributions in a single place. At resubmission of your final files, please consider providing an ORCID ID for as many contributing authors as possible.

13) JCB requires authors to submit Source Data used to generate figures containing gels and Western blots with all revised manuscripts. This Source Data consists of fully uncropped and unprocessed images for each gel/blot displayed in the main and supplemental figures. Since your paper includes cropped gel and/or blot images, please be sure to provide one Source Data file for each figure that contains gels and/or blots along with your revised manuscript files. File names for Source Data figures should be alphanumeric without any spaces or special characters (i.e., SourceDataF#, where F# refers to the associated main figure number or SourceDataFS# for those associated with Supplementary figures). The lanes of the gels/blots should be labeled as they are in the associated figure, the place where cropping was applied should be marked (with a box), and molecular weight/size standards should be labeled wherever possible.

14) Journal of Cell Biology now requires a data availability statement for all research article submissions. These statements will be published in the article directly above the Acknowledgments. The statement should address all data underlying the research presented in the manuscript. Please visit the JCB instructions for authors for guidelines and examples of statements at (<https://rupress.org/jcb/pages/editorial-policies#data-availability-statement>).

B. FINAL FILES:

Thank you for this interesting contribution, we look forward to publishing your paper in Journal of Cell Biology.

Sincerely,

Julius Brennecke, PhD
Monitoring Editor
Journal of Cell Biology

Dan Simon, PhD
Scientific Editor
Journal of Cell Biology

Reviewer #1 (Comments to the Authors (Required)):

In the revised version the authors have addressed most of my comments. However, some points remain - this might be due to misunderstandings. I do recommend language editing of the manuscript.

For example:

The authors changed the following sentence that reads now: "Tej lacking the Lotus domain (Tej- Δ Lotus) formed cytoplasmic aggregates and exhibited co-localization with Spn-E (Fig. 2B). Consistent with the findings of a previous study⁴⁵, Tej did not co-localize with Vas."

I think the authors mean: ... Tej lacking the Lotus does not co-localize with Vasa. The Ephrussi lab shows that Tejas co-localizes with Vasa but the Tejas/Vasa interactions depends on the Lotus domain.

The authors provide all five alphafold models plus the PAE plots, which suggests that the prediction is robust. However, it is unclear to me, why the authors did not include the in the supplement; Alphafold predictions are still relatively new and we should try establish high standards that allow the readers to judge the quality of the data. I would insist that these data are added to the supplement.

I brought up Tapas because it is similar to Tejas on the domain level but seems to be less important in the biological systems. Tapas can as Tejas bind Vasa, but I assume it lacks the SRS and therefore can not bind SpnE. This would be an interesting point to make in the discussion. I would appreciate such a rationale explanation why Tejas/Tapas look similar, but are different.

Concerning the phase separation part:
I still do not understand the following sentence:

Consistent with this notion, IUPred analysis predicted an intrinsically disordered region in the middle part of Tej (Fig. 5A), prompting us to examine whether Tej is employed in the propensities equivalent phase separated granule 53, 54.

Maybe? Tejas contains a predicated intrinsically disordered region prompting us to examine its contribution to condensate formation.

When I suggested: "To show that Vasa/SpnE mobility is affected by Tejas - the authors could delete the Tejas/Lotus domain or make use of the Tej mutant that does not bind SpnE anymore." I meant that what happens to vasa mobility when Tejas lacks the SRS? This would also provide insight into the interplay between Vasa/SpnE.

It would have been nice to include the deltaIDR also panel S5A - this would allow comparison of all tejas constructs side-by-side. Now the authors included the schematic construct, but it would have been nice to also include the experimental data - so that all tejas localization patterns can be directly compared in one figure.

Reviewer #3 (Comments to the Authors (Required)):

The authors have addressed my concerns well. I have no further requests and support publication in JCB!

Dear Editor,

We are submitting the final version of the manuscript titled “Tejas functions as a core component in nuage assembly and precursor processing in *Drosophila* piRNA biogenesis” (JCB manuscript #202303125). We appreciate your generous support for the constructive revision process.

As suggested by the reviewer #1, we amended text and added the AF2 models and PAE plots into Supplementary Figure 3 in the revised manuscript. Please kindly refer point-to point responses below, if needed.

We added scale bars for the magnified images and molecular weight as editor pointed out. In addition, according to the JCB author instruction, we changed red color in the labels, lines and images to magenta in those paired with green color, and shortened the abstract less than 160 words. We also added Author contribution description, and amended Data and Material Availability, Funding Statement, and Conflict of Interest Statement as well. The revised manuscript has 24,143 characters for title page, abstract, introduction, results, discussion, and acknowledgments), 6 figures, 6 supplemental figures/legends, and 3 supplemental tables. We hope that this version is suitable for publication in *Journal of Cell Biology*.

Thank you for your consideration. We look forward to hearing from you.

Sincerely,

Toshie KAI

Reviewer #1 (Comments to the Authors (Required)):

In the revised version the authors have addressed most of my comments. However, some points remain - this might be due to misunderstandings. I do recommend language editing of the manuscript.

For example:

The authors changed the following sentence that reads now: "Tej lacking the Lotus domain (Tej- Δ Lotus) formed cytoplasmic aggregates and exhibited co-localization with Spn-E (Fig. 2B). Consistent with the findings of a previous study⁴⁵, Tej did not co-localize with Vas."

I think the authors mean: ... Tej lacking the Lotus does not co-localize with Vasa. The Ephrussi lab shows that Tejas co-localizes with Vasa but the Tejas/Vasa interactions depends on the Lotus domain.

We appreciate that the reviewer pointed this out. This line was amended as suggested.

Page 6, Line 116-119; Tej lacking the Lotus domain (Tej- Δ Lotus) formed cytoplasmic aggregates and exhibited co-localization with Spn-E (Fig. 2B). In contrast, consistent with the previous finding, Tej- Δ Lotus did not co-localize with Vas, indicating that the interaction between Tej and Vas depends on the Lotus domain.

The authors provide all five alphafold models plus the PAE plots, which suggests that the prediction is robust. However, it is unclear to me, why the authors did not include the in the supplement; Alphafold predictions are still relatively new and we should try establish high standards that allow the readers to judge the quality of the data. I would insist that these data are added to the supplement.

We added them to FigS2 as requested.

I brought up Tapas because it is similar to Tejas on the domain level but seems to be less important in the biological systems. Tapas can as Tejas bind Vasa, but I assume it lacks the SRS and therefore can not bind SpnE. This would be an interesting point to make in the discussion. I would appreciate such a rationale explanation why Tejas/Tapas look similar, but are different.

We added the argument in Discussion part as requested.

Page 10, Line276-279; Another protein known as Tap, which is a fly counterpart of Tdrd7 and harbors Lotus and eTudor domains, has previously been reported to participate in the piRNA

pathway and interact with Vas. However, since Tap lacks the SRS found in Tej, it is unlikely to be involved in the recruitment of Spn-E.

Concerning the phase separation part:

I still do not understand the following sentence:

Consistent with this notion, IUPred analysis predicted an intrinsically disordered region in the middle part of Tej (Fig. 5A), prompting us to examine whether Tej is employed in the propensities equivalent phase separated granule 53, 54.

Maybe? Tejas contains a predicated intrinsically disordered region prompting us to examine its contribution to condensate formation.

We appreciate this suggestion by reviewer. Accordingly, we amended the text as follows. Page9, Line 215-217; Tej contains a predicated intrinsically disordered region (Fig. 5A), prompting us to examine its contribution to condensate formation.

When I suggested: "To show that Vasa/SpnE mobility is affected by Tejas - the authors could delete the Tejas/Lotus domain or make use of the Tej mutant that does not bind SpnE anymore." I meant that what happens to vasa mobility when Tejas lacks the SRS? This would also provide insight into the interplay between Vasa/SpnE.

We agree it would provide some insights. However, the absence of SpnE perturbs piRNA precursor processing, and hence we think this experiment would provide only inconclusive results. However, based on the observation that the mobility of Tej lacking the SRS was reduced in FigS5B, it is reasonable to speculate that the interacting Vas would also has less mobility.

It would have been nice to include the deltaIDR also panel S5A - this would allow comparison of all tejas constructs side-by-side.

Now the authors included the schematic construct, but it would have been nice to also include the experimental data - so that all tejas localization patterns can be directly compared in one figure.

We added it to Fig S5 as requested. Please note that we placed the image of mKate2-tagged Tej-ΔIDR since no GFP-fused one was available with DAPI.